# Machine learning-assisted decoding of temporal transcriptional dynamics via fluorescent timer

Nobuko Irie[1], Naoki Takeda[2], Yorifumi Satou [1], Kimi Araki [2,3] & Masahiro Ono [1,4] ✉

Investigating the temporal dynamics of gene expression is crucial for understanding gene regulation across various biological processes. Using the Fluorescent Timer protein, the Timer-of-cell-kinetics-and-activity system enables analysis of transcriptional dynamics at the single-cell level. However, the complexity of Timer fluorescence data has limited its broader application. Here, we introduce an integrative approach combining molecular biology and machine learning to elucidate *Foxp3* transcriptional dynamics through flow cytometric Timer analysis. We have developed a convolutional neural network-based method that incorporates image conversion and class-specific feature visualisation for class-specific feature identification at the single-cell level. Biologically, we developed a novel CRISPR mutant of Foxp3 fluorescent Timer reporter mice lacking the enhancer Conserved Non-coding Sequence 2, which revealed new roles of this enhancer in regulating *Foxp3* transcription frequency under specific conditions. Furthermore, analysis of wild-type *Foxp3* fluorescent Timer reporter mice at different ages uncovered distinct patterns of *Foxp3* expression from neonatal to aged mice, highlighting prominent thymus-like features of neonatal splenic *Foxp3*+ T cells. In conclusion, our study uncovers previously unrecognised *Foxp3* transcriptional dynamics, establishing a proof-of-concept for integrating CRISPR, single-cell dynamics analysis, and machine learning methods as advanced techniques to understand transcriptional dynamics in vivo.

Understanding the temporal dynamics of gene expression is fundamental to comprehending gene regulation's functional significance, cellular differentiation, and development. Single-cell level analyses of diverse tissues including differentiating cells[1,2] and developing or functioning T cells[3–5] have utilised trajectory analyses to reconstruct in vivo activation or differentiation processes. However, trajectory analysis inherently relies on similarity analysis through dimensional reduction, which is not a direct measurement of temporal dynamics in vivo. Thus, the current challenge lies in accurately capturing and measuring temporal elements of these processes within individual cells.

Fluorescent Timer proteins offer a unique solution to this challenge. Previous studies include a mathematical modelling approach to cell population dynamics[6]. However, the full integration of single-cell techniques with a Timer-based approach has yet to be fully realised. We previously developed the Timer-of-cell-kinetics-and-activity (Tocky) system for single-cell flow cytometric analysis of cellular activities and transcription[7]. Tocky uses a mutant mCherry fluorescent

[1]The Joint Research Center for Human Retrovirus Infection, Kumamoto University, Kumamoto, Japan. [2]Institute of Resource Development and Analysis, Kumamoto University, Kumamoto, Japan. [3]Center for Metabolic Regulation of Healthy Aging, Kumamoto University, Kumamoto, Japan. [4]Department of Life Sciences, Imperial College London, London, United Kingdom. ✉e-mail: m.ono@imperial.ac.uk

timer protein, Fast-FT, which spontaneously and irreversibly changes its chromophore from blue to a mCherry-type red form with the maturation half-life 4.1 h[8]. The mature red-form Timer protein is stable and its decay rate is 122 h[9]. Using this Tocky approach, we have developed Foxp3-Tocky and Nr4a3-Tocky mice, enabling us to analyse the temporal dynamics of Foxp3 transcription and transcriptional activities downstream of T cell receptor (TCR) signalling, respectively[7].

While Foxp3 has traditionally been designated as the lineage-specific transcription factor for regulatory T cells (Tregs), implying a stable and continuous expression as its default state[10], it is increasingly recognised as a dynamically regulated gene within CD4$^+$ T cells[11–13]. The Foxp3-Tocky system has significantly contributed to our understanding by uncovering the highly dynamic expression of Foxp3 during both inflammatory and homoeostatic states[9]. Key activating signals for Foxp3 include TCR, Interleukin-2 (IL-2) and transforming growth factor-beta (TGF-β). In addition, Foxp3 expression is regulated by an autoregulatory loop, where Foxp3 protein impacts its own transcription[9]. Intriguingly, the interaction between Foxp3 and Runx1 not only controls Foxp3 function[14] but also affects Foxp3 transcription[15]. Along with Stat5, NFAT[16], CREB, and Ets-1[17–19], Foxp3 and Runx1 bind to Conserved Non-Coding Sequence 2 (CNS2)[20], which is a key enhancer in the Foxp3 gene and is essential for maintaining Foxp3 expression via epigenetic modifications[16,20].

Previous studies have provided foundational insights into CNS2 and its role in sustaining Foxp3 expression post-cell division and potential roles in reactive expression in response to TCR and IL-2 signals[16,20,21]. Specifically, these studies demonstrated that, in CNS2 KO T cells, Foxp3 expression is lost as cells divide in culture[20,21], and under inflammatory conditions, Foxp3$^+$ T cells can lose Foxp3 expression[21]. However, past research has relied predominantly on bulk analyses of cell populations from CNS2 KO mice, where the CNS2 region of the endogenous Foxp3 gene is deleted. Such approaches have not pinpointed specific cells in which CNS2 is actively functioning. Thus, it remains unresolved how a functioning CNS2 controls Foxp3 transcription in real time at single-cell resolution, what types of temporal dynamics of transcription active CNS2 induces, and whether and how IL-2 and TCR signal downstream genes are induced under intact CNS2 activity. This gap in understanding underscores the need for methodologies that can dissect these temporal dynamics at a granular, single-cell level, to better elucidate the functional mechanisms of CNS2.

Meanwhile, the influence of developmental and ageing processes on Foxp3 transcription remains largely unknown. Previous studies often concentrate on thymic neonatal stages alone, without offering a comprehensive spleen-thymus comparison[22]. Compounding this issue, controversial evidence regarding Treg developmental dynamics in neonates has circulated widely, creating substantial confusions in the research field[23]. While Foxp3-expressing T cells are known to accumulate in aged individuals, the biological significance of this accumulation remains unclear[24]. Moreover, studies examining aged spleen and thymus typically involve only two time points—adult and aged[25,26]. Accordingly, our current study aims to develop standard quantitative data on the dynamic regulation of Foxp3 transcription across tissues and throughout both development and ageing.

To address our biological aims, we have developed a novel data-driven framework integrating advanced molecular biology techniques, including Foxp3-Tocky and CRISPR-mediated mutagenesis within the reporter construct of the Tocky system. Given the continuous nature of Timer fluorescence distribution[9], it is essential to eliminate manual gating methods from our research framework. Manual gating, constrained by manually pre-determined gates, is vulnerable to arbitrariness and subjectivity, leading to non-reproducibility[27], and is particularly problematic given the continuous nature of Timer Blue and Red fluorescence data. Additionally, while flow cytometry provides high-throughput single-cell analysis, its static snapshots of dynamic processes, combined with the limited dynamic range of Timer fluorescence, present significant challenges[28]. Moreover, there is no universal mathematical approach to transforming Timer fluorescence data back into transcriptional kinetics.

To overcome these limitations, we have developed a suite of machine learning (ML) methodologies specifically tailored for flow cytometric Timer analysis. These methodologies directly analyse the spatial patterns in two-dimensional flow cytometric Timer data, enabling a comprehensive capture of group-specific Timer dynamics. By eliminating the arbitrariness and subjectivity associated with manual gating and obviating the need for rigid assumptions about Foxp3 transcriptional dynamics or Timer profiles, this approach ensures that the outputs are data-driven and reproducible. The effectiveness of our methodologies is confirmed by quantitative model performance metrics, underscoring their reliability.

Supervised ML approaches including Random Forest (RF) and Convolutional Neural Networks (ConvNets) have been successfully applied to genomic data[29,30] and multidimensional flow cytometric data[31–35]. However, methods designed for multi-marker datasets can introduce bias when applied directly to Timer fluorescence data[36]. Thus, we introduce two complementary approaches: TockyKmeansRF, which integrates clustering with RF analysis, and TockyConvNet, a ConvNet framework employing a novel image conversion technique and Gradient-weighted Class Activation Mapping (Grad-CAM). This toolkit moves beyond conventional gating and unsupervised clustering in flow cytometry, enabling a more sophisticated and data-oriented analysis of Foxp3 transcriptional dynamics at the single-cell level.

## Results

### Overview of novel ML approaches to analyse flow cytometric tocky data

Figure 1a outlines the significant pitfalls and risks associated with manual gating, the most prevalent method in immunology for identifying cells of interest in cytometry analysis. The most widely used gating methods are rectangle, polygon and ellipse gates, all of which are arbitrarily hand-drawn and highly problematic. While these gates allow immunologists "flexible" identification of cell populations to test a hypothesis, they inherently introduce substantial arbitrariness and subjectivity, increasing variability and reducing transparency in data analysis[37,38]. Gates depend on predefined features of cell populations (e.g., a rectangle gate for CD25$^{high}$ Foxp3$^{high}$), embedding various hidden and ambiguous assumptions and potential biases into the analysis. Designed to isolate populations of interest to test a hypothesis, such arbitrariness makes the analytical process prone to *confirmation bias*— the tendency to cherry-pick data that supports pre-existing beliefs[29]. These arbitrary and subjective elements undermine the transparency and rigour of data analysis, contributing to the reproducibility crisis, which is widespread in life sciences and preclinical studies[39], including the research fields using flow cytometric analysis[27].

In our strategic shift from manual gating to a data-oriented approach, we have developed a new research framework that utilises ML to enable coherent two-dimensional analysis of Timer Blue and Red fluorescence, departing from conventional two-variable analysis and significantly enhancing the power of Tocky. Figure 1b depicts a workflow within this framework, aimed at unravelling transcriptional dynamics in a functional system through ML-assisted identification of group-specific features. Using the Tocky system, transcriptional dynamics influenced by an enhancer are investigated by CRISPR-induced mutation of the enhancer within the Foxp3 Timer transgene. Independent experiments are conducted to generate training and test datasets through flow cytometric analysis of the Timer fluorescence profile of T cells. ML models are then trained to classify samples into experimental groups using the Timer fluorescence data. Subsequently, model performance analysis is conducted using the test data to obtain model performance metrics and validate the trained model. The

**a**

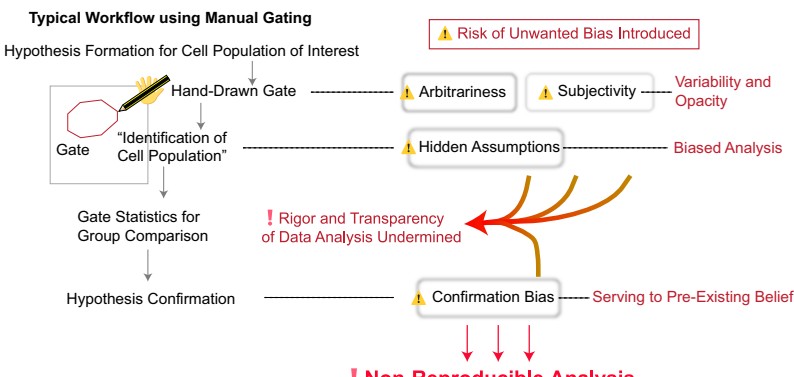

**b**

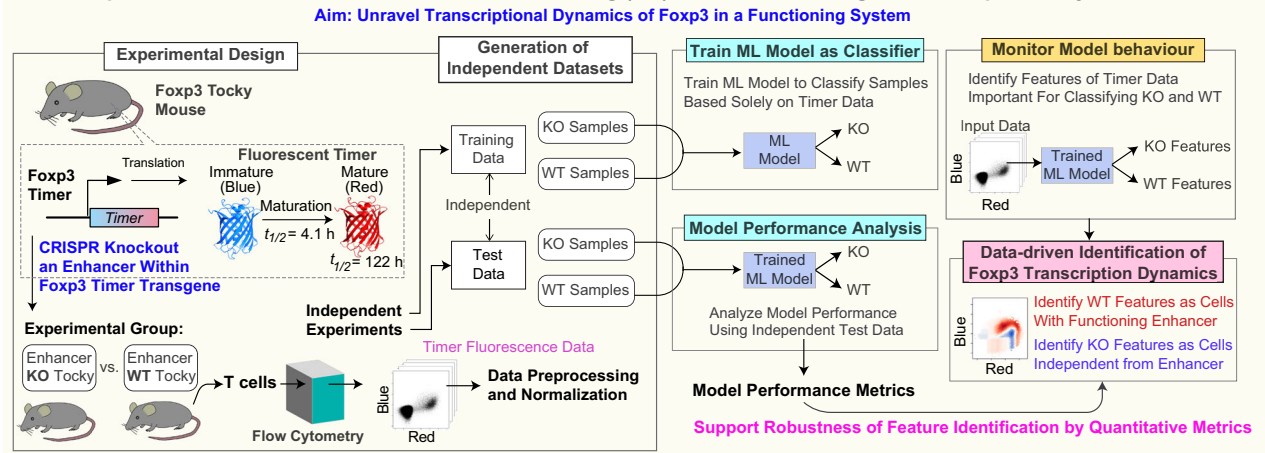

**c**

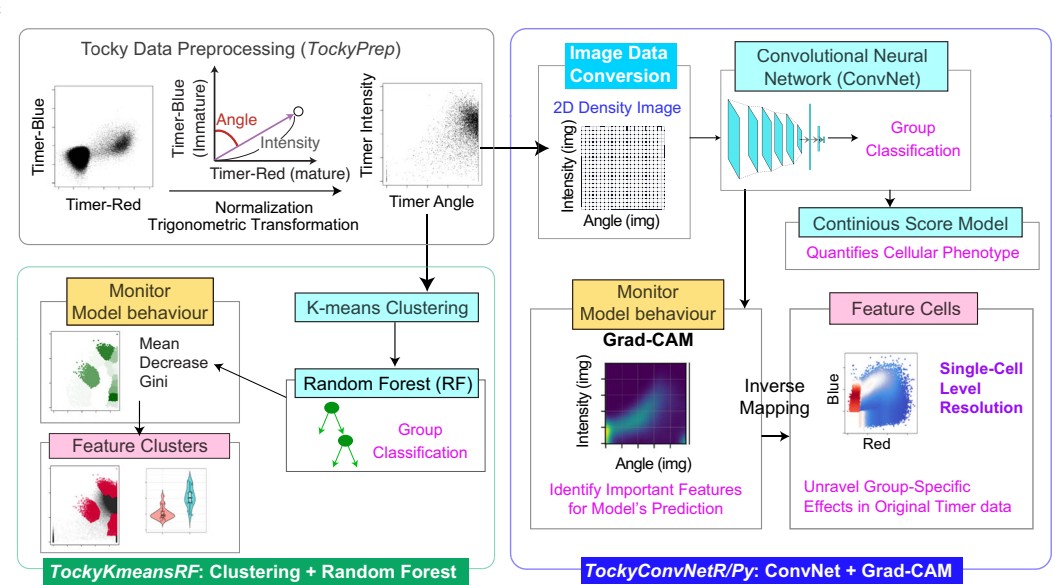

behaviour of the trained model is then analysed by tailored methods, which enables the identification of group-specific *feature cells*. These feature cells represent the group-specific *features* that assist in the classification of samples across datasets. Importantly, our approach is designed to perform cross-dataset analysis by applying the trained ML model to new data inputs to dynamically identify group-specific feature cells. The robustness of the identified group-specific features is quantitatively supported by the model performance metrics, enhancing transparency and rigour of the data analysis process.

Under this novel research framework, we developed and implemented two primary ML approaches, *TockyKmeansRF* and *TockyConvNet*, each tailored to identify feature cells within Timer fluorescence data (Fig. 1c). Both methods are used to train ML models as classifiers, which are monitored and analysed to identify group-

**Fig. 1 | Research Framework and Overview of Machine Learning Methods.**
**a** Pitfalls and risks in manual gating are schematically presented, highlighting the major pitfalls and risks associated with manual gating, emphasising the hand-drawn nature of the methodology that introduces bias and undermines reproducibility. Unicode emojis for warning and hand symbols are included[83]. **b** Proposed research framework for Machine Learning (ML)-assisted decoding of transcriptional dynamics. This schematic outlines the comprehensive workflow employed to unravel transcriptional dynamics of Foxp3 within a functional system. It covers the experimental design, generation of independent training and test datasets, training of ML models, performance evaluation, and data-driven identification of group-specific feature cells through model behaviour analysis. The protein structure is adapted from[82]. **c** Implementation of the research framework as *TockyMachineLearning*, a novel machine learning suite designed for this study. Data preprocessing, performed by *TockyPrep*, normalises and transforms flow cytometric Timer data into standardised Timer Angle and Timer Intensity data. This pre-processed data then feeds into the TockyMachineLearning toolkit. Within this toolkit, TockyKmeansRF combines k-means clustering with Random Forest (RF) analysis, utilising the mean decrease Gini index to identify feature cells. Tock-yConvNet transforms Timer Angle and Intensity data into 2D grayscale images representing cell density. These images are batch-processed by ConvNet, with model behaviours monitored using Grad-CAM to enable identification of feature cells at the single-cell level.

specific features. TockyKmeansRF integrates k-means clustering with Random Forest (RF) analysis, using the mean decrease Gini index to monitor model behaviour. Meanwhile, TockyConvNet converts this data into two-dimensional "images" for ConvNet analysis, employing Grad-CAM to identify key regions influencing network predictions[40]. In addition, we show the capability of TockyConvNet to establish a continuous scoring system for quantitatively analyse cellular phenotype.

To use these models, flow cytometric Timer Blue and Red fluorescence data are pre-processed and transformed into Timer Angle and Timer Intensity, as previously described[7] and implemented as a computational tool[41]. *Timer Angle* is measured from the y-axis, represented by Timer Blue fluorescence, towards the x-axis, represented by Timer Red fluorescence. *Timer Intensity* is the magnitude (or *norm*) of the vector formed by these fluorescence values (Fig. 1c).

## A novel experimental tool to investigate the roles of conserved non-coding sequence 2 (CNS2) in regulating temporal dynamics of Foxp3 transcription

To identify a biologically significant enhancer sequence and establish a prototypic approach to studying Foxp3 transcriptional dynamics, we analysed Chromatin Immunoprecipitation sequencing (ChIP-seq) data. Our analysis demonstrated that both Foxp3 and Runx1 proteins uniquely bound to the CNS2 region of the Foxp3 gene (Fig. 2a) as reported previously[20]. Importantly, our investigations using Foxp3-Tocky revealed that Foxp3 protein is required for sustaining Foxp3 transcription[9] and that the CNS2 region is actively demethylated at the moment when Foxp3 transcription is sustained and persistent in the CD4 single-positive thymocytes[7]. Therefore, we hypothesised that CNS2 functions as a platform for critical transcription factors, including Foxp3 itself and Runx1, to dynamically regulate the Foxp3 transcriptional activities. Deleting the CNS2 sequence should therefore elucidate the temporal phases of Foxp3 transcriptional regulation that are dependent on CNS2 (Fig. 2b).

The deletion of a sequence within a bacterial artificial chromosome (BAC) transgene could be done in vitro, followed by the creation of a new mouse strain. However, such an approach is susceptible to between-founder variations, an inherent issue in BAC reporter systems[42]. Meanwhile, modifying the endogenous Foxp3 sequence could make any output reporter measurement secondary to the modified dynamics of the Foxp3 protein[43]. Therefore, it was essential to delete the CNS2 sequence within the BAC Foxp3-Tocky transgene only, without disturbing the endogenous Foxp3 gene.

We achieved this by using a CRISPR KO method combined with a dedicated breeding strategy. Fertilised eggs from Foxp3-Tocky mice underwent CRISPR-based electroporation (Materials and Methods). To facilitate the deletion of CNS2, a single-stranded oligodeoxynucleotide carrying homology arms was used to enable homologous recombination and replace the CNS2 region with a short oligo (Fig. 2c). Critically, we established two distinct PCR assays: a CNS2 deletion-specific PCR to detect both the endogenous and BAC Foxp3-Timer loci ("*Common CNS2 Deletion*") and Foxp3 Timer-specific PCR to discriminate WT Foxp3 | Timer and CNS2 KO Foxp3 Timer ("*Foxp3 Timer-Specific*

*Discrimination PCR*", Fig. 2d). These assays identified founder mouse #87, which carried the CNS2 KO Foxp3 Timer without any evidence of CRISPR editing in the endogenous Foxp3 gene (Fig. 2e–f). Sanger sequencing further confirmed the successful deletion and homologous recombination of the CNS2 locus in the BAC transgene (Fig. 2g).

The founder mouse #87 was used to establish a breeding line through successive matings with WT mice. Over multiple generations spanning more than two years, we selectively bred progeny that expressed the CNS2 KO Foxp3 Timer, consistently backcrossing them to the B6 background. This extensive breeding and selection process confirmed that the CNS2 KO Foxp3-Tocky transgene was stably inherited in a Mendelian manner (Fig. 2h), reassuring that the modification involved a single transgene in an autosomal chromosome. Our breeding strategy and the backcrossing ensured the CNS2 deletion was specific to the Foxp3 Timer transgene, thereby effectively eliminating any possibility of CRISPR-induced alterations to the endogenous Foxp3 locus. The specificity of the Foxp3 Timer-Specific Discrimination PCR was further validated by Sanger sequencing (Fig. 2i–j). Based on these validations, we used hemizygous Foxp3-Tocky and hemizygous CNS2 KO Foxp3-Tocky mice as parents in all experiments, excluding double transgenics, and ensured that littermate analysis was consistently employed throughout the study.

## Analysis of CNS2 KO Foxp3-Tocky using established methods

After establishing the CRISPR mutant strain, we first examined the effects of CNS2 deletion with conventional flow cytometric analyses (Fig. 2k). Mean fluorescence intensity (MFI) measurements revealed only a moderate, albeit significant, decrease in Timer Red fluorescence in KO T cells ($p < 0.01$), with Timer Blue largely unchanged (Fig. 2l). We next applied a trigonometric transformation[7,44], converting Timer fluorescence into Timer Angle and Intensity. Although the Timer Locus categorisation method[44] indicated that active Foxp3 transcription, identified as *NPt*, *Persistent*, and *PAt* categories, was reduced in KO T cells (Fig. 2m), these existing approaches rely on predominantly one-dimensional analyses and failed to capture the nuanced dynamics of CNS2-mediated Foxp3 transcription. This limitation underscored the need for more comprehensive methods to dissect Foxp3 transcriptional regulation.

## TockyKmeansRF: Clustering and random forest analysis of timer fluorescence

Figure 3a illustrates the TockyKmeansRF implementation, a combinatorial ML method that integrates k-means clustering with RF classification[45]. TockyKmeansRF constructs an RF model using training flow cytometric Timer fluorescence data, which is subsequently tested on an independent test dataset. Initially, TockyKmeansRF applies k-means clustering separately to both training and test datasets, producing two tables that show the percentage of cells in each cluster. Clusters between these datasets are matched based on the Euclidean distances between them (Materials and Methods). The cluster percentage table from the training set is then utilised to build an RF model, which is evaluated using the corresponding table from the test

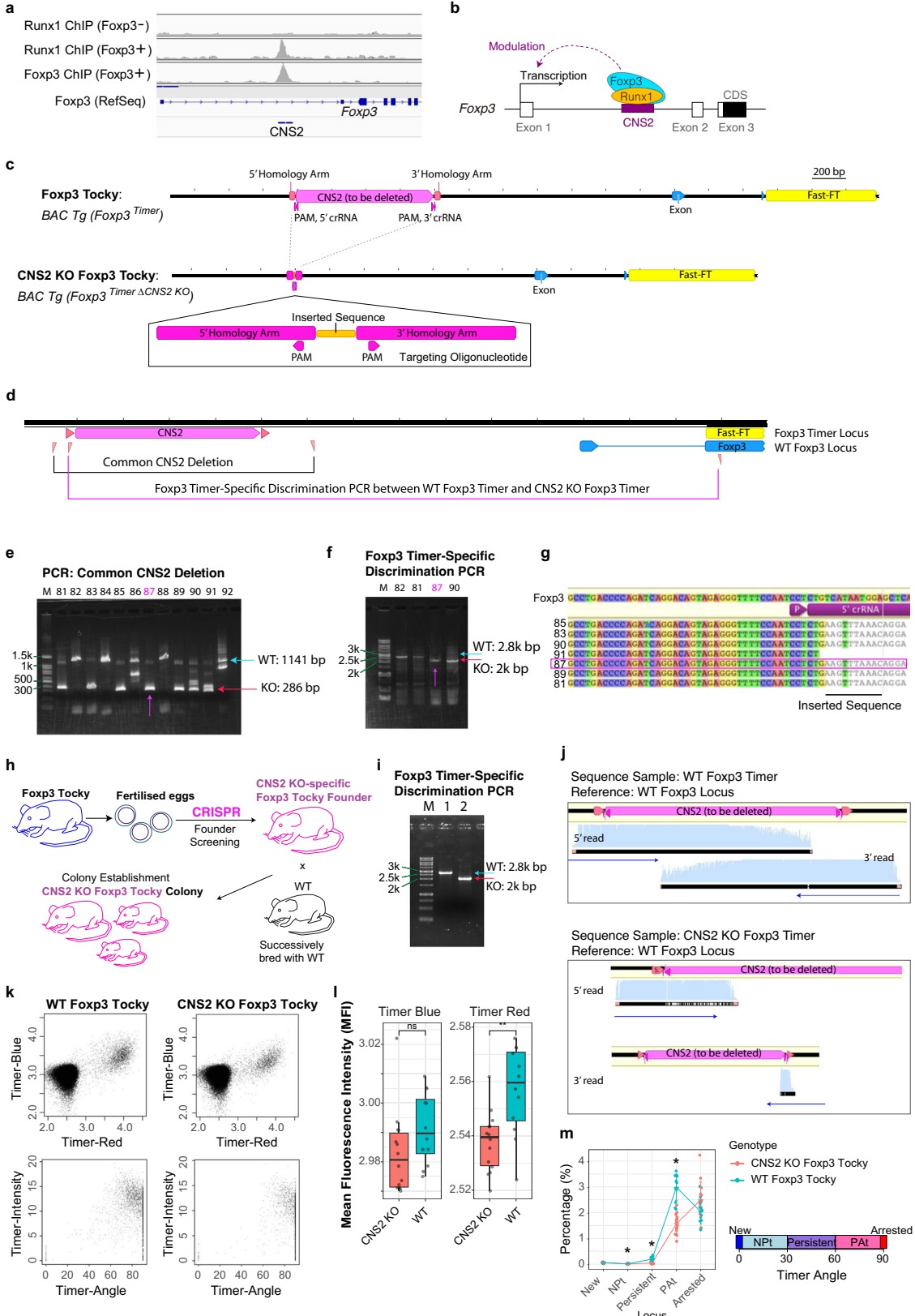

set. The mean decrease Gini (MDG) index helps identify significant clusters and, consequently, feature cells within the original Tocky data[46].

To validate the effectiveness of the ML approaches in the current study, we generated two independent datasets by conducting flow cytometric analysis on lymph node samples from WT Foxp3 Tocky and CNS2 KO Foxp3 Tocky littermates. This process produced the initial training and test datasets discussed in the following sections (Fig. 3b, Supplementary Table 1).

First, we assessed the robustness of the TockyKmeansRF model by varying the number of clusters in k-means clustering and the number of trees in the Random Forest model. We confirmed stable

**Fig. 2 | Development of the CRISPR-mediated CNS2 KO Foxp3-Tocky Mouse Model. a** ChIP-seq profiles showing aligned sequence reads for Runx1 and Foxp3 binding in Foxp3+ and Foxp3− CD4+ T cells. **b** Working model of CNS2-dependent Foxp3 transcriptional regulation by Runx1 and Foxp3. **c** CRISPR-Cas9 strategy for CNS2 deletion within the Foxp3-Timer locus, showing the targeting oligonucleotide design. **d** Two PCR-based genotyping approaches: one detects CNS2 deletion in both endogenous and transgenic Foxp3 loci; the other discriminates WT and CNS2 KO Foxp3-Timer alleles. Screening of founder mice using **e** Common CNS2 Deletion PCR and **f** Foxp3 Timer-specific PCR. Screening was performed once during founder selection following embryonic injections. **g** Sanger sequencing confirming deletion in founder mice. **h** Breeding strategy illustrating how CNS2 KO Foxp3-Tocky mice were successively bred with WT animals over several generations to establish a stable mouse colony while ensuring that the endogenous *Foxp3* gene remains free from mutations. **i** Established genotyping PCR for discriminating CNS2 KO Foxp3-Tocky and WT Foxp3-Tocky mice;

representative result from >3 independent tests. **j** Sanger sequencing to validate specificity of the Foxp3 Timer-specific PCR shown in **h**. **k** Representative flow cytometric plots of Timer Blue vs. Timer Red and Timer Angle vs. Intensity for WT and CNS2 KO Foxp3-Tocky mice. **l** Box plots showing MFI of Timer Blue and Timer Red fluorescence in CD4+ T cells. The box represents the interquartile range (IQR; 25th–75th percentile), showing the median, and whiskers up to 1.5× IQR from the box. Statistical significance was assessed using a two-sided Student's *t*-test ($p < 0.01$). Exact *p*-values: Timer Red = 0.004; Timer Blue = 0.053. $n = 14$ KO and 12 WT samples. **m** Percentage of cells within each Timer locus in CD4 + T cells from superficial lymph nodes. Locus definitions: New (0˚), NPt (0˚–30˚), Persistent (30˚–60˚), PAt (60˚–90˚), Arrested (90˚). Error bars show standard deviations. Two-sided Mann–Whitney test with *p*-value adjustment used. Adjusted *p*-values for Tocky loci: Persistent = $2.1 \times 10^{-6}$, PAt = $6.2 \times 10^{-6}$, Arrested = $1.3 \times 10^{-2}$. $n = 14$ KO and 12 WT samples.

performance across a range of cluster and tree numbers by conducting area under the curve (AUC) analysis (Fig. 3c). The TockyKmeansRF model demonstrated a commendable classification accuracy. Constructed using training data with 18 clusters and subsequently tested on an independent dataset, the model achieved an out-of-bag (OOB) error rate of 7.69%. The confusion matrix from the test dataset indicates a high predictive accuracy, with an overall accuracy of 91.18%. This performance underscores the model's robustness and its capability to distinguish effectively between the KO and WT classes within the testing framework, supporting the significance of the feature cells as follows.

Using the MDG index as an importance score, CNS2-dependent feature cells were identified among individual single cells within the test dataset, specifically in the Timer Angle and Intensity space and within the original Timer fluorescence data (Fig. 3d). Density-based clustering of these feature cells revealed three distinct clusters (Fig. 3e). Cluster 1 was predominantly found in KO mice, whereas Clusters 2 and 3 were more prevalent in WT mice in the training data (Fig. 3f). Notably, cells in Cluster 2 from WT Foxp3 Tocky mice exhibited high expression levels of CD25 (Interleukin-2 receptor alpha chain) and PD-1. Meanwhile, cells in Cluster 3 showed increased expression of CD69 and CD44 (Fig. 3g). Thus, each cluster had a unique activation profile and CNS2 KO T cells shifted from Clusters 2 and 3 to Cluster 1, markedly reducing Timer Intensity and approaching to the Timer Angle 90, which indicates the arrested transcription[7].

Computational performance metrics, such as runtime and memory usage, were evaluated during the execution of TockyKmeansRF on the scaled CNS2 KO dataset (Fig. 3h). During training, the maximum runtime was approximately 3 s per training session, and peak memory usage did not exceed 30 MB, even as the sample size increased to 136. Notably, the number of trees in the RF model did not significantly impact either runtime or memory usage.

These findings underscore the efficacy of TockyKmeansRF in identifying unique CNS2-dependent feature cells and classifying CNS2 KO-specific patterns. This demonstrates the model's robustness and precision in analysing complex dynamics of Timer fluorescence profiles, which are crucial for establishing a data-oriented approach to studying temporal transcriptional dynamics using the Tocky system.

## TockyConvNet: A ConvNet approach using Grad-CAM for discriminating WT and CNS2 KO Foxp3 Timer

The successful deployment of TockyKmeansRF, coupled with the optimisation of a relatively high number of clusters, indicates that transforming Timer fluorescence data into image data could open new avenues for ML applications, particularly by leveraging ConvNet technologies. The conversion process involved binning the data into $100 \times 100$-pixel images (Fig. 4a), effectively preserving the essential visual characteristics of the Timer data after conversion (Fig. 4b).

To prevent overfitting, for TockyConvNet we designed a compact ConvNet model consisting of two convolutional layers. Each layer features a sigmoid-activated pointwise convolution, termed "Spatial Attention", enhancing spatially relevant feature extraction (Materials and Methods). The architecture includes two dense layers (Fig. 4c). The model successfully learnt through three-fold cross-validation using a relatively small number of training epochs (Fig. 4d).

The model's efficacy was validated through ROC analysis on an independent test dataset. Furthermore, we benchmarked the TockyConvNet model against traditional manual gating methods employed in Fluorescent Timer analysis. These manual methods include Quadrant gating for distinguishing Timer Blue and Red positivity, and Polygons for cells with high Blue levels above and below the diagonal line between Timer Blue and Red, *Polygon-Blue(high)* and *Polygon-Red(high)*, respectively (Supplementary fig. 1). TockyConvNet achieved excellent performance metrics, with both the Area Under the Curve (AUC) of the ROC and Average Precision scoring 1.0. In contrast, manual gating methods demonstrated significantly lower performance: the polygon gate scored 0.87 for AUC and 0.76 for Average Precision, while the quadrant gate scored 0.5 for AUC and 0.41 for Average Precision (Fig. 4e–f).

## Grad-CAM analysis of CNS2-Dependent Foxp3 timer dynamics

To elucidate CNS2-dependent Foxp3 transcription dynamics, Grad-CAM was applied across various convolutional layers to visualise transcriptional features that distinguish two genotypes. Precisely, for each layer, gradients for each pixel across all feature maps were calculated, globally averaged, and weighted. These weighted feature maps were then transformed into a single heatmap through pixel-wise summation, retaining only positive activations using the ReLU function (Materials and Methods).

To analyse the features of CNS2-dependency effectively, we generated differential heatmaps from Grad-CAM outputs of WT and KO samples using each convolutional layer of our model (Fig. 4g). These heatmaps identified pixels critical for classification, illustrating how Grad-CAM progressively reveals these pixels across convolutional layers. The outputs were then reverse-mapped to their respective positions in the Timer Angle-Intensity space and to their original locations in the raw Timer Blue and Red fluorescence space (Fig. 4h).

Quantitative comparisons between the two genotypes were conducted by analyzing cells in the top 90th percentile as WT CNS2 feature cells (CNS2-dependent), and those in the bottom 10th percentile as KO feature cells (CNS2-independent). This analysis revealed that while all convolutional layers discerned differences between the two Foxp3 Tocky variants, the most pronounced differences were observed in the last convolutional layer, Attention2-Conv (Fig. 4i). In contrast, the cells increased in the KO group were predominantly captured by the first three convolutional layers (Fig. 4j).

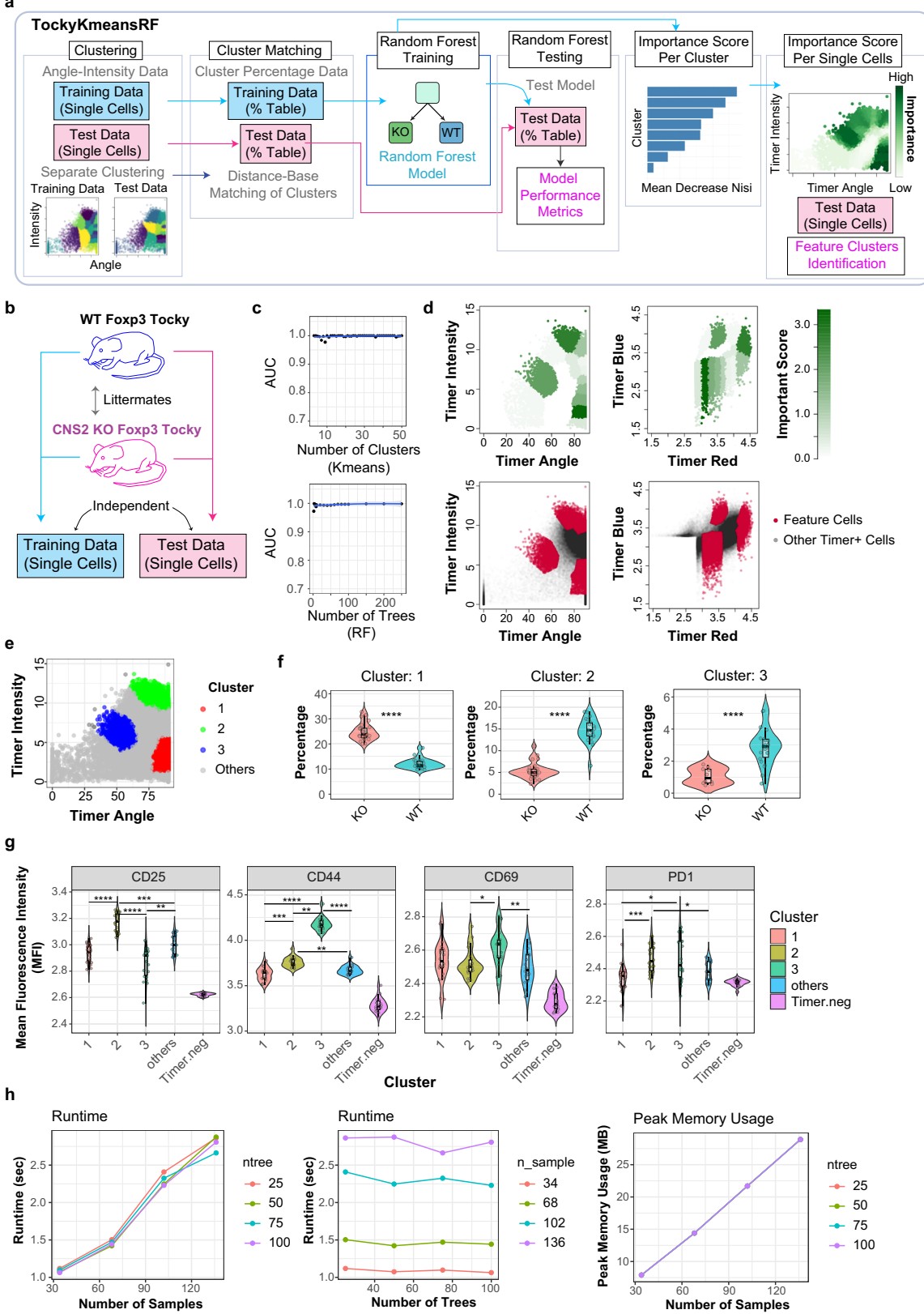

The CNS2-dependent feature cells identified across all convolutional layers, as shown in Fig. 4i, exhibited higher CD44 expression compared to other Timer+ cells (Fig. 4k). Notably, the cells highlighted by the Attention2-Conv layer displayed significantly elevated expression of CD69 and PD1, distinguishing them from other Timer+ cells.

## Gene regulation in CNS2-Dependent feature cells

To further explore the biological significance of CNS2-dependent feature cells in relation to the temporal regulation of Foxp3 transcription, we analysed RNA-seq data from flow-sorted Timer-positive cells from WT Foxp3 Tocky mice, fractionated into B1, B2, R1, and R2 fractions[9] (Fig. 5a). Utilising *TockyPrep* for data preprocessing[41], we

**Fig. 3 | TockyKmeansRF: Combinatorial analysis using clustering and random forest. a** Schematic overview of the TockyKmeansRF framework combining k-means clustering and Random Forest (RF) classification to model Timer fluorescence in flow cytometry data. **b** Training and test datasets generated from lymph node samples of WT and CNS2 KO Foxp3-Tocky mice. **c** Area Under the Curve (AUC) analysis of model performance across varying numbers of clusters (top) and RF trees (bottom). **d** Visualisation of CNS2-dependent feature clusters in the test dataset. Timer Angle–Intensity and original Timer fluorescence spaces are shown. Top: feature importance by Mean Decrease Gini (MDG); bottom: feature cells defined as top 60th percentile by MDG. **e** Density-based clustering of feature cells. **f** Violin plots showing kernel density estimates of the percentage of cells in each cluster per sample ($n$ = 22 KO, 27 WT). Each point represents a biological replicate. The box indicates the interquartile range (IQR; 25th–75th percentile), the centre

line denotes the median, and whiskers extend to the most extreme values within 1.5× IQR; outliers beyond this are plotted. Samples lacking cells in Cluster 3 (KO) were excluded from that cluster's plot. **g** Violin plots showing mean fluorescence intensity (MFI) of CD25, CD44, PD-1, and CD69 for two identified clusters and remaining Timer[+] cells ("others") in WT samples ($n$ = 27). Statistical analysis used the Kruskal-Wallis test followed by Dunn's test with Bonferroni correction. Timer-negative cells were included only as a baseline reference and excluded from statistical testing. The box shows the IQR, the centre line indicates the median, and whiskers extend to the most extreme values within 1.5× IQR. Exact $p$-values are provided in Supplementary Data 1. **h** Computational performance of TockyKmeansRF, showing runtime and memory usage with progressively increased CNS2 KO training data.

converted Timer fluorescence data from these sorted cells into Timer Angle and Intensity values. This conversion facilitated the application of Grad-CAM analyses in Fig. 4g to dynamically identify CNS2-dependent and independent cells as CNS2 WT and KO feature cells, based solely on Timer distribution within the flow cytometric data linked to RNA-seq data.

Precisely, using the extensive Grad-CAM analyses shown in Figs. 4g – j, CNS2-dependent cells were identified using *WT Feature* of Attention-Conv2, while CNS2-independent, inactive cells were pinpointed using *KO Feature* of Conv2 (Fig. 5b). This cross-dataset analysis revealed that the fraction B2 was highly enriched with CNS2-dependent cells, comprising over 40% of the cells in this fraction, while the other fractions contained only a few such cells (Fig. 5c). Conversely, CNS2-independent cells were predominantly found in the fraction R2, representing over 50% of the cells, while more than 30% in the fraction R1 as well.

The CNS2-dependent fraction B2 were characterised by uniquely high expression of NFAT genes (Nfatc1 and Nfatc2), distinctly among TCR signal downstream genes. In contrast, the CNS2-independent fractions R1 and R2 highly expressed other TCR signal downstream genes including Egr1, Nr4a1, Nr4a3, Rel, and Rela, but notably not the NFAT genes (Fig. 5e). These findings suggest that each fraction is associated with unique TCR signal dynamics, which may be also influenced by additional signalling pathways. Importantly, Foxp3 expression was highest in fraction B2, aligning with the high-frequency transcriptional dynamics observed by Foxp3 Tocky. In addition, CNS2-dependent B2 cells also showed elevated expression of genes associated with Foxp3 function, such as Tnfrsf4, Tnfrsf18, Ctla4, Icos, and the TGF-β receptor component Tgfbr1. Intriguingly, the dynamics of IL-2 signal-related genes in the B2 fraction showed a distinctive pattern, with upregulation of IL-2 receptors (Il2ra and Il2rb) and repression of Stat5a (Fig. 5f).

Collectively, our results confirm that CNS2-dependent cells exhibit the highest Foxp3 expression, aligned with the highest-frequency of Foxp3 transcription revealed by Tocky (Figs. 4c, 5b, and g). In addition, the gene expression profile of CNS2-dependent cells supports that CNS2 orchestrates Foxp3 transcription under finely tuned and unique TCR signalling dynamics predominantly mediated by NFAT. The downstream activities of TCR signalling notably exclude other well-characterised genes downstream of TCR signalling such as NF-κB and Nr4a genes and are potentially influenced by unique dynamics of IL-2 signalling as well (Figs. 4c and 5g).

### Expanding the application of TockyConvNet to understand developmental and ageing Foxp3 transcription dynamics

Having demonstrated the utility of TockyConvNet with the CNS2 KO datasets, we aimed to further generalise the ConvNet method by generating and analysing independent flow cytometric datasets using Foxp3-Tocky mice. To this end, we analysed CD4[+] T cells from the spleen and the thymus of WT Foxp3-Tocky mice across various ages,

including neonates and aged mice, to capture the full spectrum of Foxp3 Timer dynamics throughout the mouse lifespan.

We discovered notable variations in Foxp3 Timer profiles, influenced by age and organ (Fig. 6a). Thymic T cells displayed new and active Foxp3 transcription, which decreased over time, resembling splenic Timer profiles in older mice. Particularly in neonates at days 3 and 4 post-birth, we observed high levels of new Foxp3 transcription in both thymus and spleen. On days 1 and 2 post-birth, the thymus included substantial numbers of CD4-single positive cells, whereas the spleen had too few CD4 + T cells to permit meaningful analysis (Supplementary Table 1b).

The flow cytometric Foxp3 Timer data were normalised and transformed into Timer Angle and Intensity formats to quantitatively analyse the datasets (Fig. 6b, c), revealing dynamic and gradual changes between the tissues across different ages.

Notably, as mice aged, Timer Blue fluorescence in both splenic and thymic Foxp3 transcription diminished, while Timer expressing cells accumulated, particularly those with low Blue and high Red fluorescence (Figs. 6a, b) with high Timer Angles (Fig. 6c).

Thus, we generated independent training and test datasets (designated as *Foxp3 Neonatal-to-Ageing Benchmarking Data*). The aim of the analysis was to understand the tissue-specific and age-dependent dynamics of Foxp3 Timer profiles. Traditional manual gating methods, including quadrant and polygon gates (Supplementary Fig. 1), along with mean Timer Angle, captured some aspects of these dynamics, especially when the age scale is transformed logarithmically (Fig. 6d, e). These observations support significant changes in Foxp3 transcription dynamics from early life into old age (Fig. 6d, e). In aged mice, Timer[+] cells accumulated, reflecting the impact of ageing on Foxp3 transcription.

These nuanced, continuous, and dynamic changes in Timer profiles across a broad range of samples make them ideal targets for analysis using TockyConvNet. To further validate this ConvNet method, we generated a training dataset and an independent test dataset, both consisting of splenic and thymic T cells from mice of various ages (Supplementary Table 1).

### Development of the TockyConvNet for quantitative assessment of thymus and spleen characteristics in Foxp3 transcriptional dynamics

First, we aimed to develop a TockyConvNet model that captures the features of thymic Foxp3 dynamics and enables classification of samples given the age of mice. Accordingly, we adapted the TockyConvNet model to classify spleen and thymus samples by including age as an input for model training (Fig. 7a). This model effectively learnt the training dataset using three-fold cross-validation and showed high performance metrics including Area under curve of ROC 0.9 and Average Precision 0.95 (Fig. 7b). Furthermore, by transferring the learned parameters from all layers and removing the softmax activation from the final dense layer, we constructed a continuous score

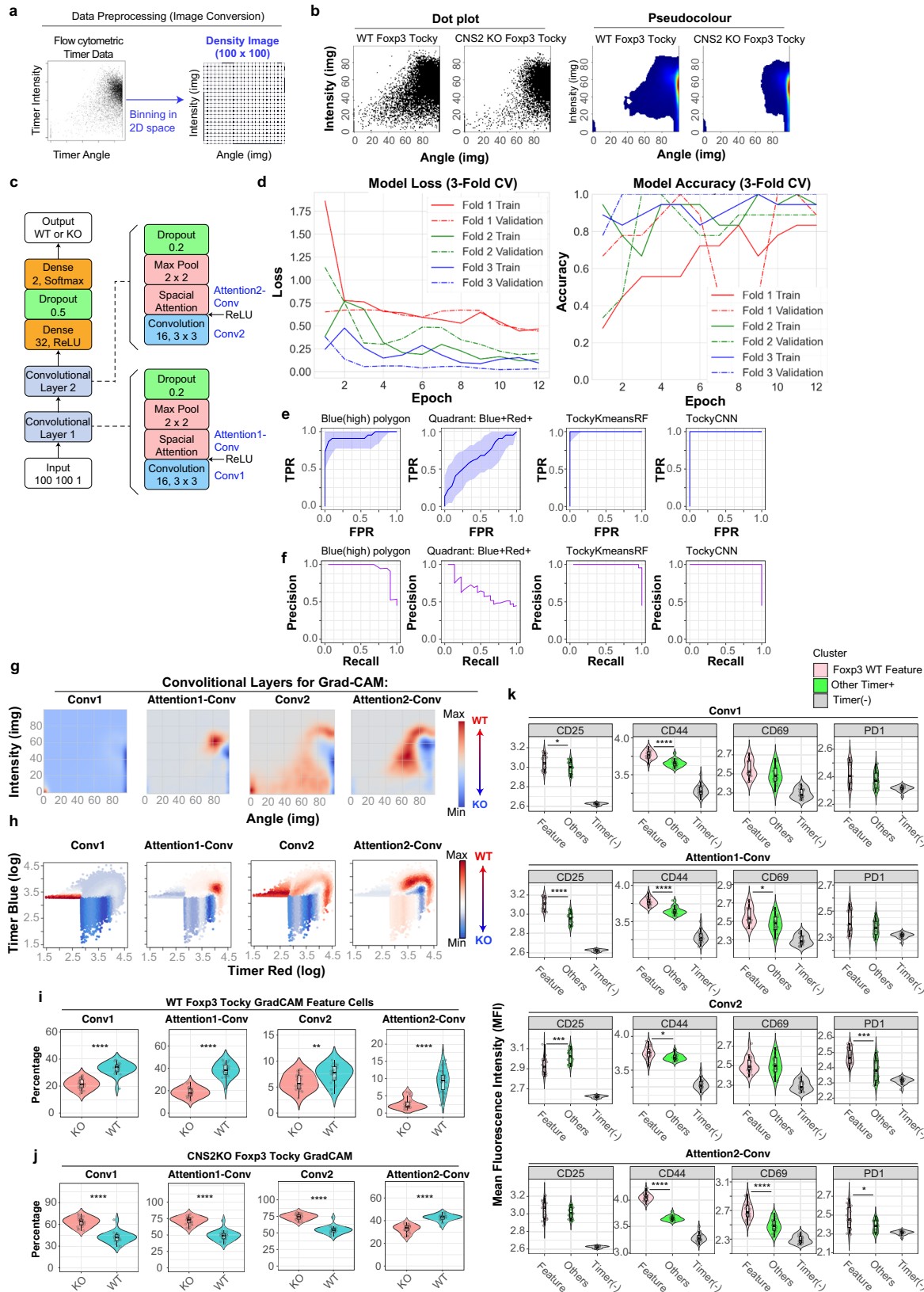

model that captures the smooth and continuous dynamics of Foxp3 transcription (Thymus-Spleen model score, Fig. 7c).

The Thymus-Spleen model score data fit better with a quadratic regression using logged age rather than raw age (Fig. 7c, Supplementary Table 2), suggesting that changes are more pronounced in early developmental stages and diminish in adult mice. Interestingly, splenic

T cells from neonatal mice, particularly at 3–4 days postpartum, showed high Thymus-Spleen model scores, comparable to those of adult thymic T cells, indicating that their Timer profiles closely resemble those found in thymic T cells. In contrast, thymic T cells from aged mice, especially those older than 10 weeks, demonstrated Timer profiles more characteristic of spleen-like T cells (Fig. 7c). These

**Fig. 4 | TockyConvNet: Deep learning-based analysis via image conversion and gradient mapping. a** Schematic of the image conversion process applied to Timer fluorescence data. **b** Representative dot plots (left) and pseudocolour images (right) after conversion. **c** Architecture of the TockyConvNet model, comprising four convolutional layers used for Gradient-weighted Class Activation Mapping (Grad-CAM) shown in **g, h. d** Learning curve from three-fold cross-validation. Receiver Operator Characteristics (ROC) **e** and Precision–Recall **f** analyses for benchmarking TockyConvNet with manual gating strategies. Differential Grad-CAM heatmaps for WT vs. CNS2 KO samples across convolutional layers, shown in Timer Angle-Intensity **g** and Timer Blue-Red **h** spaces. Colour range is normalised per panel. Violin plots showing kernel density estimates of CNS2 feature cell percentages in WT (**i**; top 90th percentile) and KO (**j**; bottom 10th percentile) based on

differential Grad-CAM maps. Each point represents a biological replicate. Statistical analysis used the two-sided Mann–Whitney test; **$p < 0.01$, ****$p < 0.001$. The box shows the interquartile range (25th–75th percentile), centre line indicates the median, and whiskers extend to the most extreme values within 1.5× IQR. $n = 22$ KO and 27 WT samples. Exact $p$-values are in Supplementary Data 1. **k** Violin plots of mean fluorescence intensity (MFI) for indicated markers in WT feature cells, other Timer-positive and -negative cells in WT samples ($n = 27$). Statistical significance was assessed using the Kruskal-Wallis test with Dunn's post-hoc test (Bonferroni correction); **$p < 0.01$, ***$p < 0.005$, ****$p < 0.001$. Timer-negative cells were included for reference only and not in statistical testing. Box and whisker definitions as above. Exact $p$-values are in Supplementary Data 1.

---

dynamics, reflecting gradual changes observed in two-dimensional plots of the raw data (Fig. 6), illustrate the model's capability to capture the spectrum of changes from neonatal stages through to and ageing process.

## Optimising and benchmarking of the TockyConvNet four-class classifier

The successful development of the TockyConvNet classifier and the continuous model scoring system demonstrates that specific patterns of Timer fluorescence dynamics are associated with Foxp3 transcriptional dynamics in both the spleen and thymus, adjusted for the age of mice. This is particularly noteworthy as the thymus in adult and aged mice, typically over 6-7 weeks old[47], may include peripheral T cells that have recirculated into the organ. Despite this, the model has successfully classified aged thymus samples correctly, confirming that T cells within the thymus of aged mice exhibit distinctive Foxp3 transcriptional dynamics, distinct from both young thymic T cells and aged splenic T cells. To comprehensively capture the compositions and real-time transcriptional dynamics across the lifetime of mice in these two major immunological organs—and to fully utilise this resource to benchmark the TockyConvNet approach—we extended the TockyConvNet approach from two-class classifier into four-class-classifier categorising samples by both organ type (thymus and spleen) and age (young, <30 days postpartum; aged, ≥30 days), aligned with standard definition of young adult thymus[48]. Consequently, the benchmarking dataset has been biologically optimised and exhibits balanced class distribution for ML modelling and analysis (Supplementary Table 1).

We assessed three distinct ConvNet architectures varying in the number of convolutional layer blocks (one, two, or three), resulting in the development of Conv1-Layer, Conv2-Layers, and Conv3-Layers TockyConvNet 4-Classifiers, respectively (Supplementary Figs. 2a–c and 3a, b). Each convolutional layer block consists of a 3×3 convolutional layer followed by a 1×1 convolutional layer serving as a spatial attention mechanism, similar to the structure used in the TockyConvNet model for CNS2 KO Foxp3 Tocky data (Supplementary Fig. 2 and Fig. 8a, Materials and Methods section). Among these, the Conv3-Layers model (Fig. 8a) demonstrated superior performance, as evidenced by ROC and Precision-Recall analyses (Fig. 8b). While the Conv2-Layers model exhibited comparable, albeit slightly reduced performance, the Conv1-Layer model showed a marked decrease in effectiveness.

Next, we examined the impact of data preprocessing on model performance. Converting Timer Fluorescence into Timer Angle and Intensity was crucial as raw fluorescence data was challenging for the ConvNet to process, leading to suboptimal performance (Supplementary Fig. 4a). Additionally, experimenting with different data resolutions, we found that neither low (25 × 25) nor high (400 × 400) resolution grids were effective (Supplementary Fig. 5a). This suggests that a resolution of 100 ×100 strikes the optimal balance by maintaining sufficient detail for accurate feature extraction and ensuring a

good density of cells across pixels, which is critical for reducing meaningless variability in the data.

Finally, we benchmarked the TockyConvNet models against conventional manual gating methods, specifically polygon and quadrant gates, which have been widely utilised in prior studies using Fluorescent Timer proteins (Fig. 8c, Supplementary Fig. 1)[49,50]. These traditional methods yielded satisfactory results for categorising Thymus-Young and Spleen-Aged samples, yet they faltered in accurately classifying the other two classes. Although the TockyKmeansRF method demonstrated notable efficiency with Thymus-Young and Spleen-Aged samples, it was less effective for Spleen-Young. Notably, TockyConvNet consistently outperformed all the other methods in both ROC and Precision-Recall analyses (Fig. 8b), establishing the TockyConvNet approach as a robust classifier using image-converted Timer data.

## Optimising Grad-CAM method for analysing Foxp3 timer dynamics

To further explore and develop the Grad-CAM method for flow cytometric Timer data, we analysed each convolutional layer's output by Grad-CAM using the optimised TockyConvNet model (Fig. 9). Heatmaps of the Grad-CAM outputs illustrate these distinctions, showing the progression of feature capture across layers (Fig. 9a). ROC analysis identified the efficacy of each convolutional layer in differentiating the four classes. The Thymus-Young and Spleen-Aged classes were distinctly recognised throughout the layers, while and Thymus-Aged and Spleen-Aged classes were the most distinctly recognised in the Conv2 layer (Fig. 9b).

Visualisation of the most informative convolutional layers in both Timer Angle and Intensity space, as well as the original Timer fluorescence space, provided further insights into important features in the transcriptional dynamics of different classes with single-cell granularity (Fig. 9c). Notably, Grad-CAM high cells in the Spleen-Aged class were predominantly located on the lower edge within the Timer Blue-Red space, with high Timer Intensity within the high 80°–90° Timer Angle range (Fig. 9c), mirroring the marked accumulation of cells within Blue-Red+ (Fig. 6e). This suggests their substantially attenuated and infrequent Foxp3 transcription after culminating Foxp3 proteins. In contrast, Grad-CAM high cells in the Spleen-Young class included both new Timer expression (Timer Blue+Red-, Angle = 0 °) and low Timer Intensity within the ~90° Timer Angle arrested class, suggesting that some spleen cells newly and moderately express Timer protein but may rapidly transition to arrested transcription states without sustained Foxp3 transcription. Meanwhile, the Thymus classes are both characterised by newly induced Foxp3 transcription, with Thymus-Young cells predominantly featured by remarkably high new transcription to intermediate Timer Angles with high Timer Intensity (Fig. 9c). Thymus-Aged cells are characterised by both new expression and high Timer Angle cells with high Timer Intensity, suggesting their ability to activate Foxp3 transcription as well as the accumulation of spleen-like cells, presumably due to the recirculation of peripheral T cells in aged mice[47].

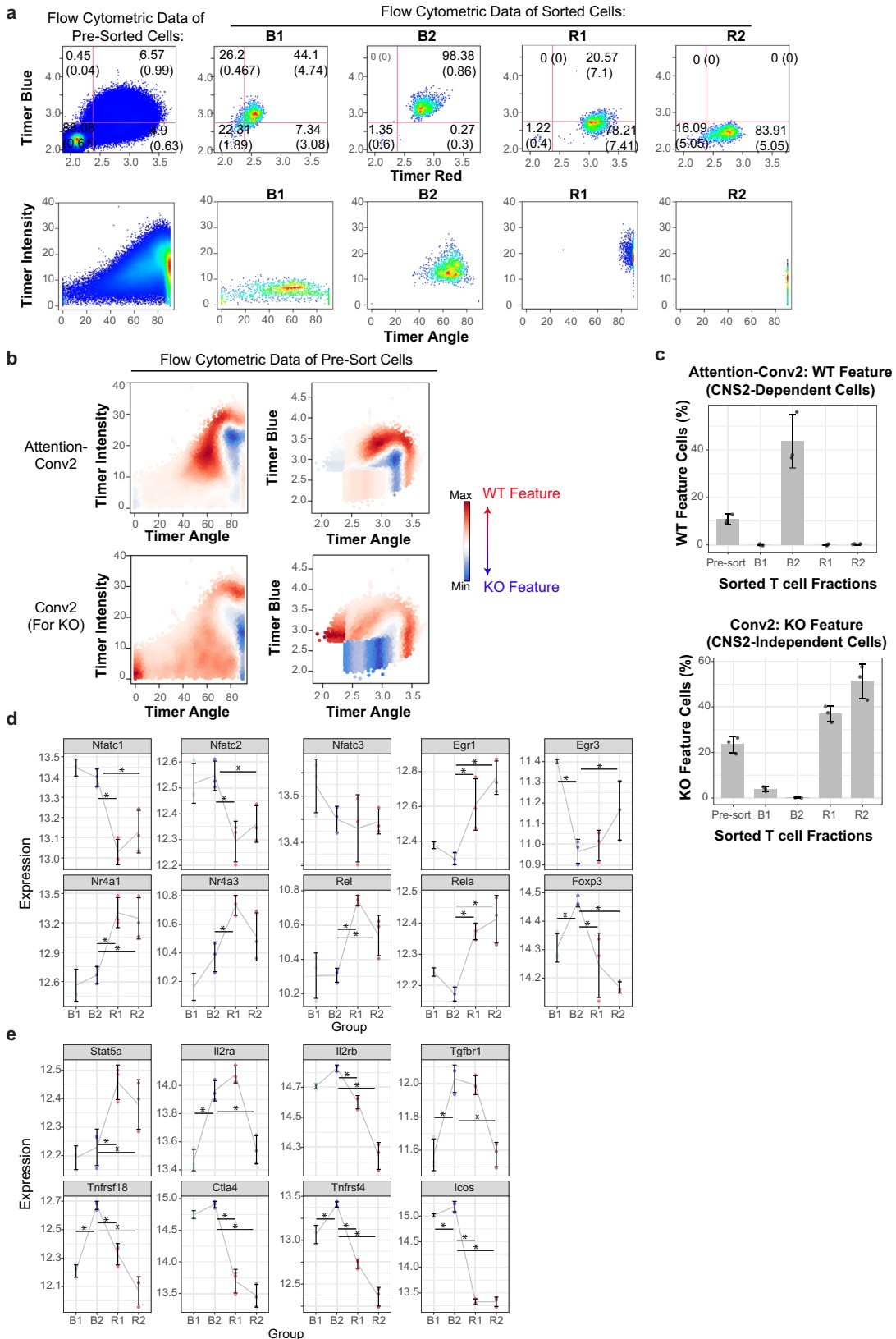

## Runtime and memory usage for TockyConvNet model learning

Lastly, the computational efficiency of TockyConvNet model learning was evaluated by analysing the runtime and memory usage. For the CNS2 KO Foxp3 Tocky data, processed with the established Conv 2-Layer model, the runtime ranged from approximately 1 to 6 s, and memory usage spanned from about 2 GB using sample sizes between 52 and 416 (Supplementary fig. 6). Similarly, using the Developmental and Ageing WT Foxp3 Tocky dataset to train the Conv 3-Layers model, runtime and memory usage also increased with larger sample sizes, ranging from around 10 to 70 s for runtime and significantly higher memory usage from approximately 1.5 to 3 GB. Together with the model complexity information in Supplementary Table 3, the analysis

**Fig. 5 | Gene Expression Analysis of CNS2-Dependent Feature Cells by Cross-Analysis of Grad-CAM Output from TockyConvNet and RNA-seq Data. a** Timer expression profile (Upper) and Timer Angle and Intensity profile (Lower) of pre-sort CD4[+] T cells and fractionated Foxp3 Timer[+] cells from WT Foxp3 Timer mice. **b** Grad-CAM heatmap using the TockyConvNet, trained as shown in Fig. 4, applied to RNA-seq flow cytometric data. (Upper) Visualisation of WT feature analysed via Attention-Conv2 Grad-CAM, highlighted by red on heatmap (Upper); (Lower) KO feature cells visualised through Conv2 Grad-CAM, highlighted with a blue on heatmap. Colour range is normalised per panel. **c** Bar charts showing the percentage of WT feature cells (i.e. CNS2-dependent cells, left) and KO feature cells (i.e. CNS2-independent cells, right) in pre-sort CD4[+] T cells and fractionated Foxp3 Timer[+] cells. Error bars indicate standard deviations. *n* = 3 biological replicates. Expression dynamics of key genes in fractionated Foxp3 Timer[+] cells: **d** transcription factors downstream of TCR signalling; **e** genes associated with IL-2 and TGF-β signalling, along with prototypic upstream and downstream Foxp3 genes. *p*-values were obtained by two-sided Wald tests of the R package DESeq2 and adjusted by the Benjamini & Hochberg method. Asterisks indicate statistical significance (adjusted *p*-value < 0.05) and shown for comparisons involving B2 only. Error bars indicate standard deviations. *n* = 3 biological replicates. Exact *p*-values are provided in Supplementary Data 1.

highlights the scalability of the TockyConvNet model, albeit with increasing computational demands for larger datasets.

## Discussion

This study establishes a proof-of-concept for dissecting physiological transcriptional dynamics at the single-cell level, introducing two principal ML methods—TockyKmeansRF and TockyConvNet—that represent a significant departure from traditional manual gating approaches. While conventional gating typically confirms known cell populations using fixed strategies, our ML approaches dynamically identify *feature cells*, which represent group-specific effects crucial for classification, based solely on the patterns of flow cytometric Timer data. TockyKmeansRF combines clustering and RF algorithms for feature-based cell identification but remains inherently cluster-focused. In contrast, TockyConvNet employs ConvNet to achieve single-cell resolution and uses Grad-CAM to provide visually intuitive, quantitative insights. Surpassing cluster-level outputs, TockyConvNet captures the finest details of Timer dynamics, enabling the precise identification of CNS2-dependent cells, as well as developmentally or ageing-specific transcriptional features of Foxp3 at the single-cell level. Benchmarking through two independent experiments robustly validates the performance and generalisability of TockyConvNet. Although TockyConvNet utilises relatively compact ConvNet models to reduce the risk of overfitting, the extensive parameters within ConvNets still pose a risk, which can be mitigated by employing shallow learning strategies and generating high-quality training data. A summary of each method's strengths and limitations is presented in Supplementary Table 4.

The Tocky ML approaches developed here are anticipated to be broadly applicable to genes beyond Foxp3 by effectively analysing Fluorescent Timer fluorescence data, facilitated by the normalisation and data transformation methods provided by TockyPrep[41]. By removing the effects of autofluorescence, these methods enable the effective application of the ML methods across various Fluorescent Timer systems (Supplementary Fig. 4). However, it is important to acknowledge that adjustments in model architecture and training methods may be necessary to accommodate different data structures, as broadly recognised within ML communities[51]. Additionally, the image conversion techniques employed in our approaches could be effectively adapted for other transcriptional reporter systems, such as EGFP, when combined with an appropriate temporal marker. This adaptation would allow the use of the TockyConvNet algorithms, which can provide a more nuanced understanding of dynamic transcriptional events. Nevertheless, such adaptations would likely require extensive experimentation and optimisation to tailor the approach to specific requirements of individual cases.

ConvNet models jointly learn features across all layers, with no single layer solely responsible for the model's learning[51]. Originally established for image classification with photographic images, Grad-CAM typically utilises the last convolutional layer, which captures abstract and semantic meanings[40]. However, our application of Grad-CAM to flow cytometric Timer data suggests that it may be necessary to investigate each convolutional layer to determine the most effective

one for summarising the features of group-specific cells. The areas highlighted by Grad-CAM likely represent the most distinct features of cells within a given class, associating high Grad-CAM values with class-specific transcriptional dynamics. This approach opens new avenues for further investigating transcriptional dynamics using Grad-CAM outputs in future studies using Foxp3 Tocky and other Fluorescent Timer reporter systems.

Biologically, by developing cutting-edge technologies such as CRISPR-based modification of only the reporter, Timer-based single-cell resolution, and machine-learning-driven data analysis, we demonstrate uniquely high-frequency Foxp3 transcriptional dynamics under functioning CNS2. This is evidenced by cells displaying low Timer Angles (~40–80°) with high Timer Intensities, which indicate persistent, high-frequency transcription[7]. This discovery challenges the traditional view that CNS2's primary role is merely to maintain Foxp3 expression post-cell division. Previous studies, relying on end-point measurements of bulk populations from endogenous CNS2 KO mice[16,20,21], were unable to differentiate between direct CNS2 activity and secondary effects from reduced Foxp3 protein. Although a previous report hinted the link between low Foxp3 expression and CNS2 activity by demonstrating loss of Foxp3 expression in relative lower Foxp3 expressors in CNS2 KO[21], all preceding studies failed to pinpoint specific cells where CNS2 was actively functioning. In contrast, our use of both WT and CNS2 KO Foxp3 Tocky mice, which retain the intact endogenous Foxp3 gene, has allowed us to identify specific CNS2-functioning cells and analyse how functioning CNS2 directly controls Foxp3 transcription dynamics, addressing a key aspect of Foxp3 autoregulation[9]. The integration of the innovative CRISPR strategy with sophisticated ML methods, through comparative analysis of CNS2 WT and KO features, confidently identify CNS2 functioning and independent cells from WT Foxp3 Tocky, revealing that intact CNS2 regulates the frequency of Foxp3 transcription bursts within specific CNS2-functioning cells. This paradigm shift, unachievable with traditional methods, is robustly supported by our high-performance ML models, unravelling a previously unrecognised temporal dimension of CNS2 function at the single-cell level.

To further obtain insights into the discovery in the temporal dimension of Foxp3 transcription, our ML-driven analysis using TockyConvNet revealed a nuanced transcriptional signature of CNS2-dependent cells, characterised by elevated IL-2R expression, repressed Stat5, and selective modulation of TCR signalling components—including increased Nfatc1/2, decreased Egr1/3, and relatively low or absent induction of Nr4a and NF-kB (identified as the B2 fraction in Fig. 5). This pattern indicates that CNS2 operates under finely tuned TCR and IL-2 signal dynamics. Such transient signalling might involve *periodic* IL-2 and TCR inputs. Consistent with our previous work employing Nr4a3 Tocky, which indicated *periodic* and *brief* TCR signals approximately every week in Foxp3-expressing cells[7], these findings imply CNS2-mediated Foxp3 transcription is regulated through intermittent and short signals rather than constant stimulation. Similarly, in vivo supply of IL-2 signalling is likely sparse in the body as each T cell can produce it for a short time only[52]. Thus, although it is well known that IL-2 enhances Foxp3 expression, its physiological transcription via

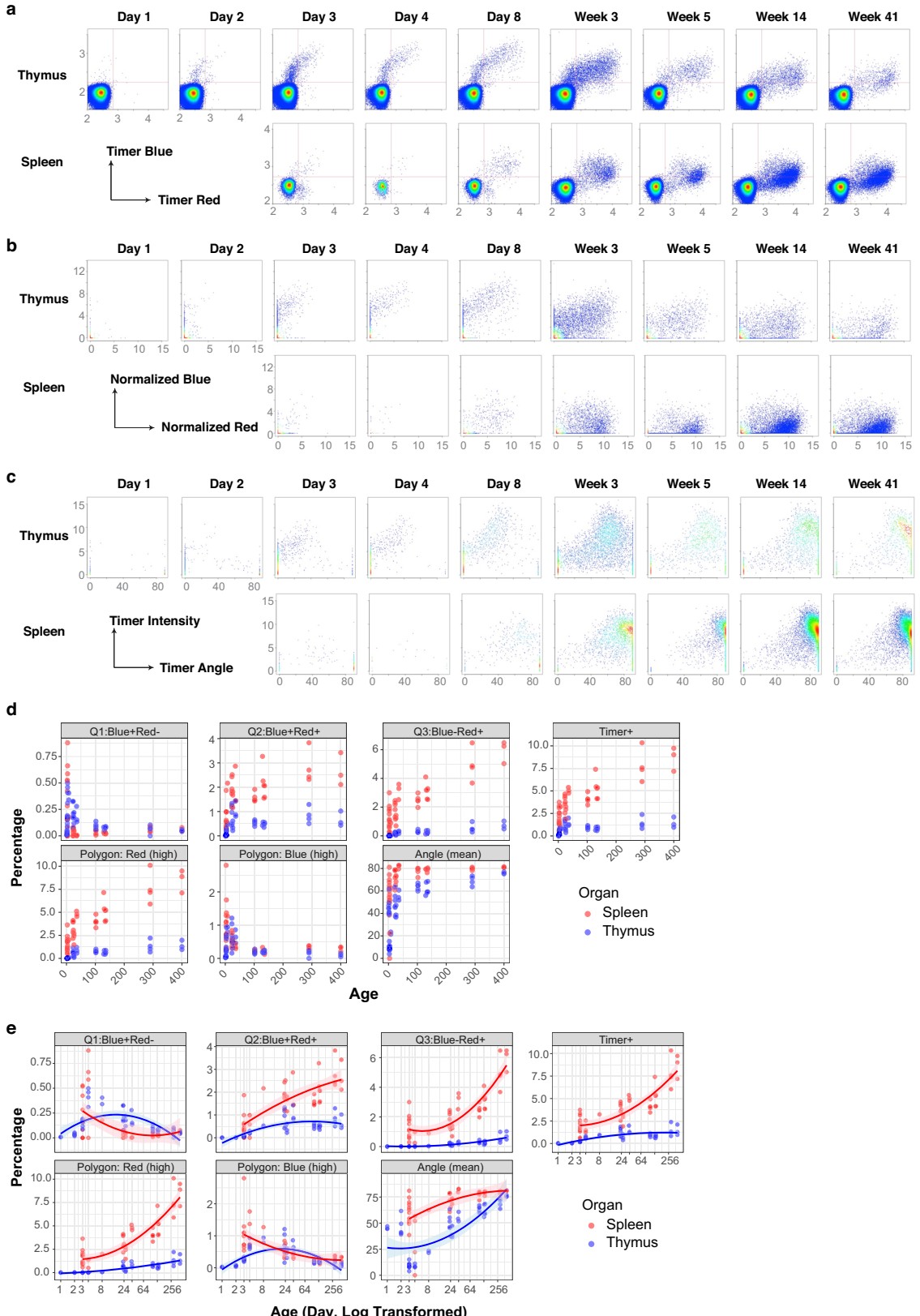

**Fig. 6 | Generation of Foxp3 Timer Neonatal-to-Ageing Benchmarking Data across developmental and ageing stages in WT Foxp3-Tocky mice. a** Timer Blue and Timer Red expression in CD4 + T cells from the thymus and spleen of WT Foxp3 Timer mice at various ages. All samples within the training dataset were concatenated per group and shown as pseudocolour plots. Days and weeks since birth are indicated. **b** Normalised Timer fluorescence data from the flow cytometric analysis in **a**. **c** Timer Angle and Intensity transformed from the normalised Timer fluorescence data in **b**. **d** Percentage of CD4[+] T cells in each of the indicated gates or the mean Timer Angle. **e** Percentage of mean Timer Angle plotted against logarithmically transformed age, with axis labels indicating actual age in days. Each line represents a quadratic regression model for each organ's data. Shaded areas indicate 95% confidence intervals around the fitted regression lines.

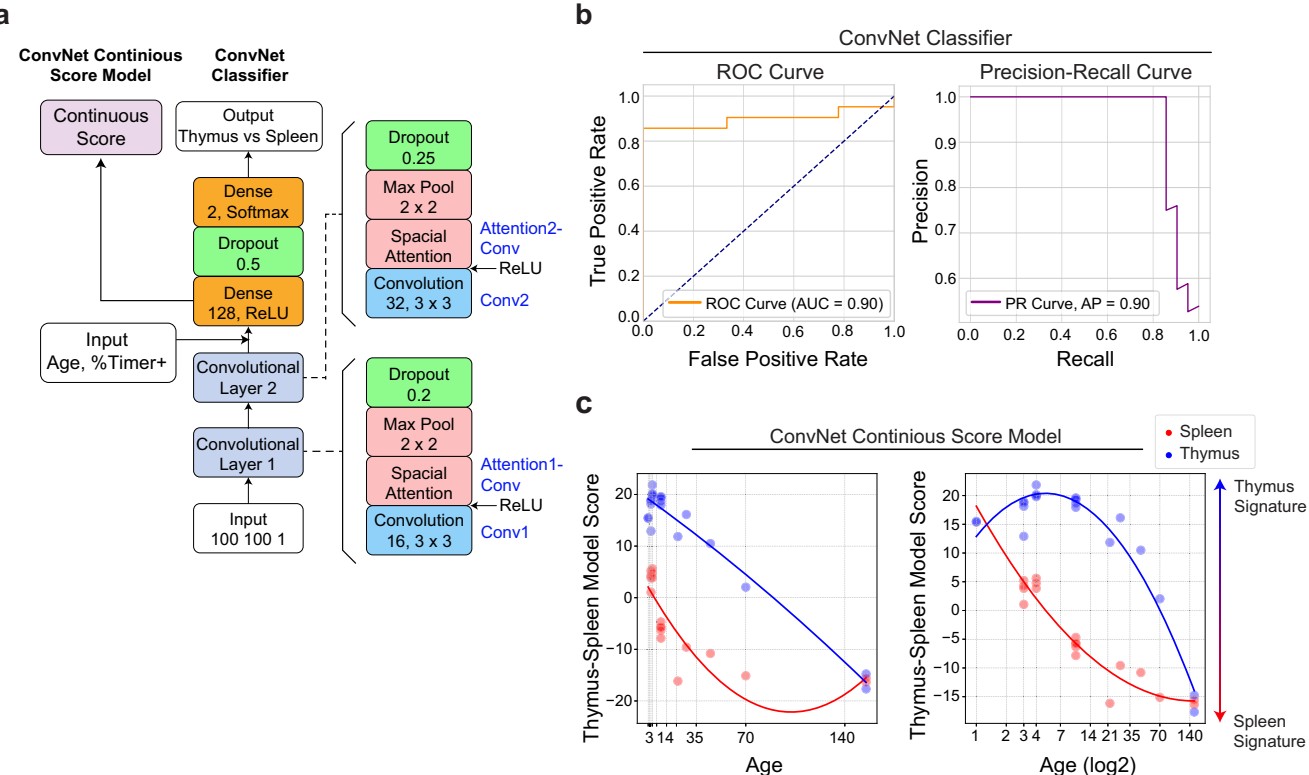

**Fig. 7 | Development of the TockyConvNet for Quantitative Assessment of Thymus and Spleen Characteristics in Foxp3 Transcriptional Dynamics.** **a** Diagram of the ConvNet architecture for the age-adjusted TockyConvNet classifier and continuous score models. **b** ROC and Precision-Recall curve analysis using the TockyConvNet classifier. **c** Thymus-Spleen Continuous Score data from the TockyConvNet continuous score model analysing an independent test dataset, using linear age values (left) and log2-transformed age values (right).

endogenous IL-2 remains incompletely understood. With our compelling evidence and the established Tocky and ML tools, future studies on Foxp3 transcriptional regulation can focus on identifying the finely tuned signalling milieu involving antigen stimulation (TCR) and IL-2 supply and signalling.

Integrating the findings, we propose a model where CNS2 controls the frequency of Foxp3 transcriptional activity through integrating transient and periodic signals from IL-2 and TCR inputs, inducing transient bursts of transcription under certain restricted situations. Future studies can build on our experimental and ML tools to further explore how the frequency Foxp3 transcription is controlled by active CNS2 – for example, this may involve integrated analysis of single-cell level methylation for CpGs in CNS2, and/or single-cell level chromatin configuration analysis, combined with our Tocky and ML technologies. Doing so, fragmented pieces of evidence regarding Foxp3 transcriptional regulation may well be integrated into a coherent and dynamic perspective. For example, CNS2 itself is regulated by Foxp3 protein[9,13,20] and downstream factors from IL-2 signalling (Stat5) and TCR signalling pathways (including NFAT[16], CREB[17] and Ets-1[18,19]). Supporting this, our flow cytometric analysis shows upregulation of PD-1 and CD25 specifically in CNS2-dependent cells. CD25 induction results from TCR and/or IL-2 signalling[53] and is further amplified by Foxp3[14], whereas PD-1 induction via TCR serves as negative feedback[54].

Further, our study has established that Foxp3 transcription is dynamically modulated by ageing, producing benchmark data for the developmental and ageing dynamics of Foxp3 transcription. In addition to the generation of the resource data, through TockyConvNet and Grad-CAM, we uncovered substantial age-dependent variations in Foxp3 transcriptional dynamics across the two major immunological organs, the spleen and the thymus.

Younger thymic T cells exhibit strong induction of new Foxp3 expression, the dynamics and trajectories of which have been captured by Grad-CAM (Fig. 9c). Intriguingly, the thymic pattern of Foxp3 transcription dynamics, as captured by the TockyConvNet continuous model, peaks at days 3-4 post-birth (Fig. 7c). This observation resonates with the uniqueness of the thymic environment in the first days of life in mice, which potentially influences negative selection processes[55].

In contrast, older thymic T cells, though still capable of initiating new Foxp3 expression, primarily accumulate cells with arrested transcription, which likely represents recirculated aged peripheral Foxp3+ T cells[47]. Recirculation of peripheral Foxp3+ T cells into the thymus typically starts only after 6 to 7 weeks post-birth[47], and thus, the cut-off of young and aged organs at 30 days old in our study effectively eliminate the contribution of recirculating T cells in young thymus. An intriguing future direction is to distinguish the dynamics of Foxp3 transcription between newly generated thymic T cells and peripheral T cells that have recirculated into the thymus in adult and aged mice, thereby addressing the possible impact of the thymic environment on recirculating T cells. Currently, the experimental identification of recirculating T cells is challenging due to limitations in available methods, even with a reporter mouse strain such as Rag2p-EGFP. These methods identify recirculating T cells as EGFP(-) cells, but critically, the expression of EGFP can be prematurely lost in newly generated thymic T cells following intrathymic cell divisions before emigration[56,57]. Therefore, future studies would benefit from combining our integrated approach involving Tocky and ML. Specifically, the development of a novel Rag2 Tocky strain, which reports Rag2 gene activity with the Fluorescent Timer protein, is a promising target. The integrated ML analysis of Timer Blue with Timer Red using our

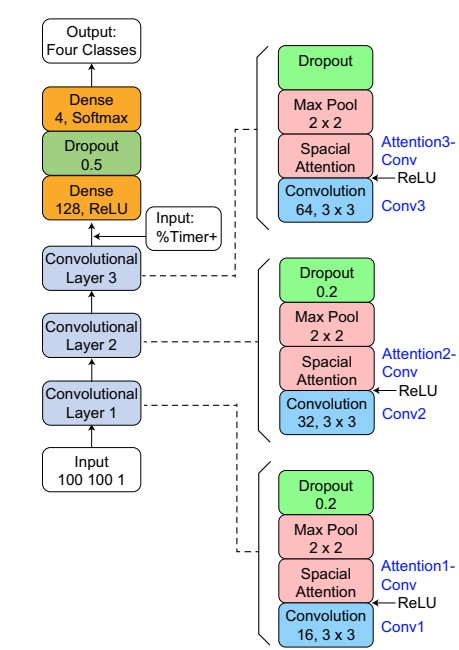

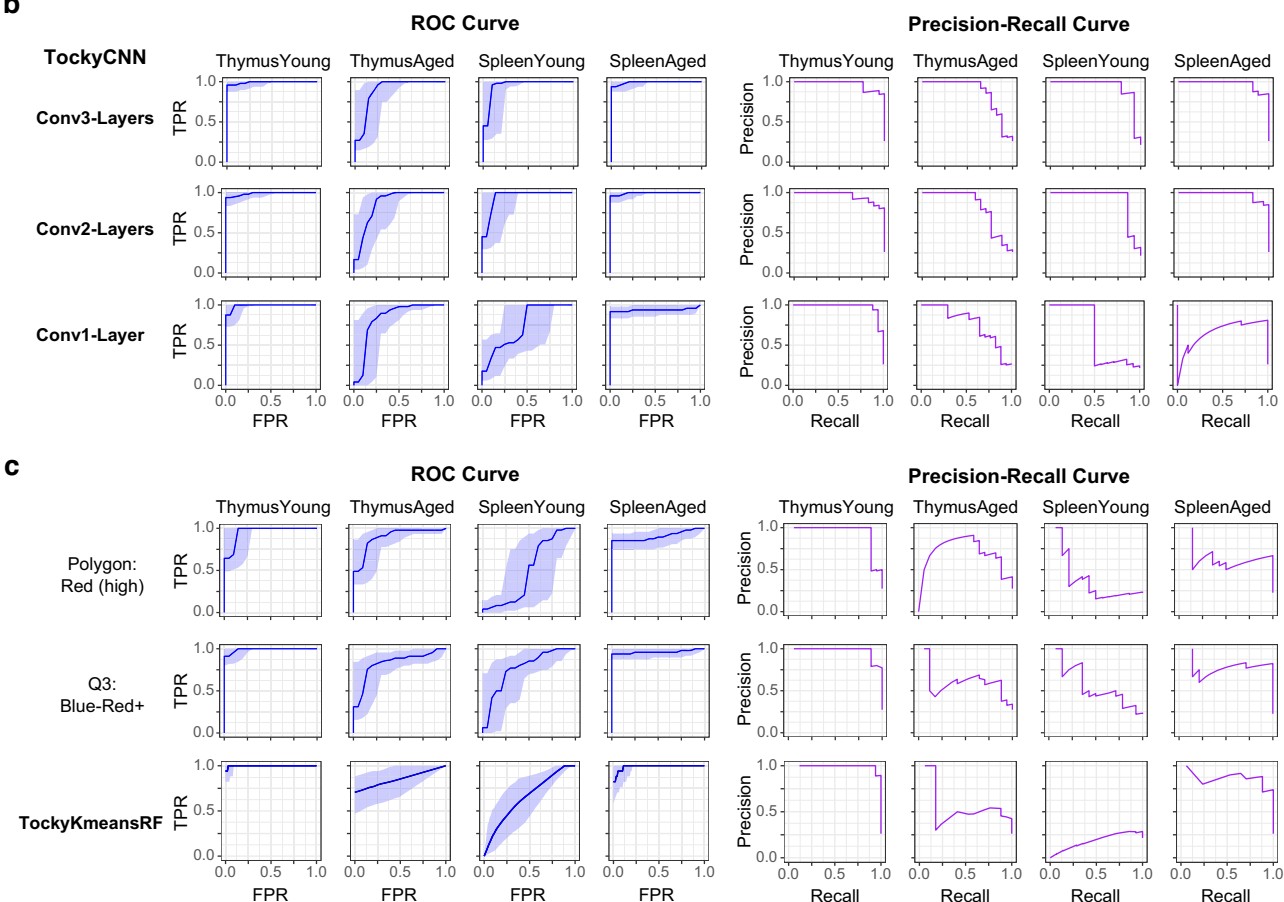

**Fig. 8 | Optimising and Benchmarking of the TockyConvNet Four-Class Classifier. a** Diagram of the 3 Conv-Layer TockyConvNet architecture as a four-class classifier for classifying the two organs (Spleen vs. Thymus) and stratified ages (Young [<30 days old]; vs. Aged (>30 days old)). **b** Optimisation of the TockyConvNet architecture, comparing models with one, two, and three convolutional layer blocks by ROC and Precision-Recall curve analysis. **c** Benchmarking results of TockyConvNet against other methods including TockyKmeansRF and manual gating strategies in differentiating the four classes using ROC (left) and Precision-Recall curves (right). The data generated in Fig. 6 were used.

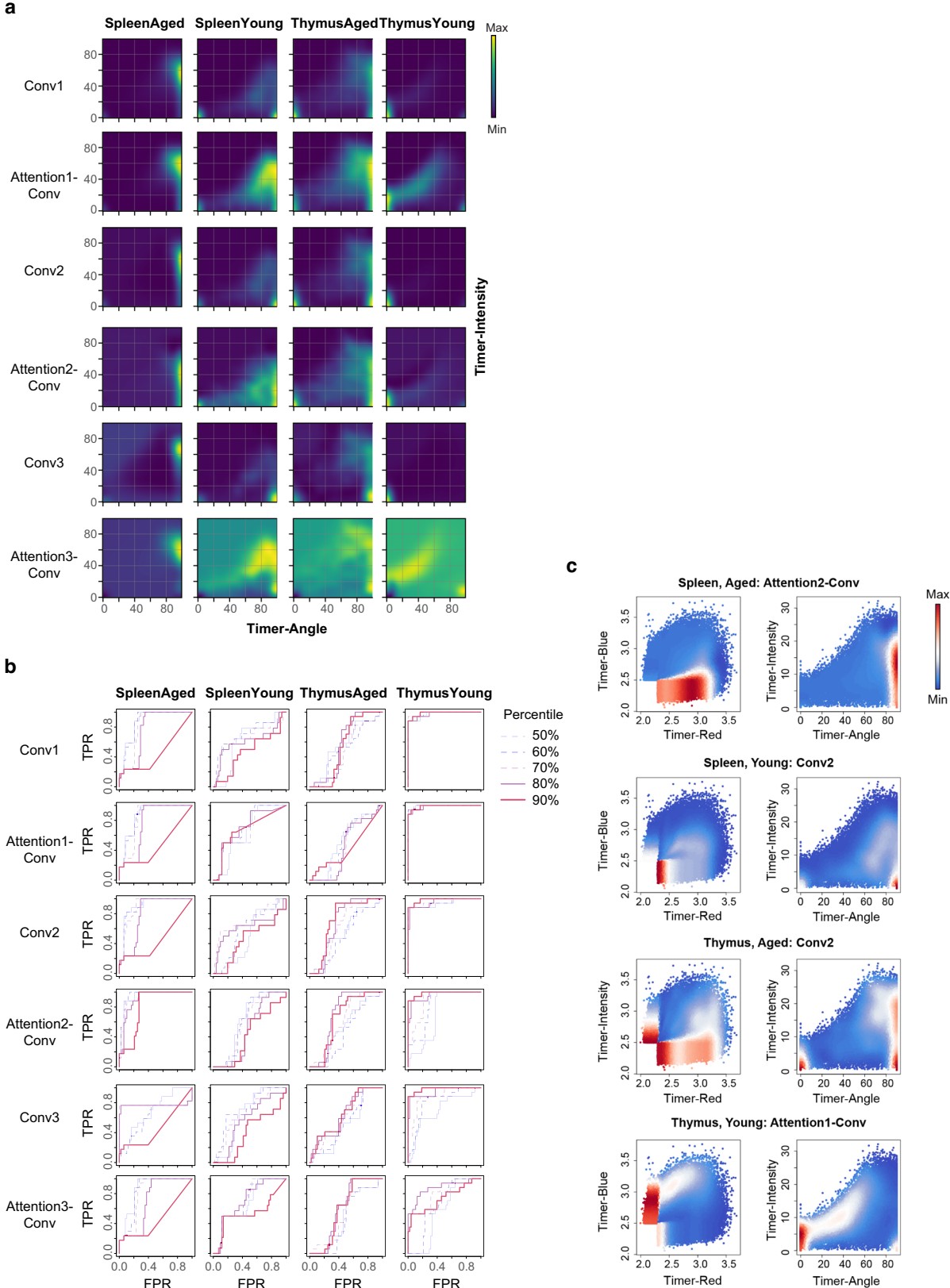

**Fig. 9 | Optimising Grad-CAM method for analysing Foxp3 timer dynamics.**
**a** Heatmaps of Grad-CAM for each convolutional layer, visualising pixels used for discriminating each indicated class. Colour range is normalised per panel. **b** ROC curves showing class-discriminative capacity of each of the convolutional layers by Grad-CAM. **c** Heatmap visualisation of the Grad-CAM outputs for each class in both Timer Angle and Intensity space, and original Timer fluorescence space. Colour range is normalised per panel.

methods should comprehensively reveal the dynamics of recently generated thymic T cells as well as recirculating T cells, thus overcoming the limitations of Rag2p-EGFP. This concept for Rag2 Tocky was initially proposed in our earlier work on Tocky[7] and the current study has renewed the relevance of this promising avenue.

In addition, intriguingly, we discovered that neonatal splenic T cells (3–4 days post-birth) actively initiate new Foxp3 expression, making their Foxp3 Timer profiles resembling to adult thymic T cells, as demonstrated by the continuous score model (Fig. 7c). Grad-CAM analysis revealed low Timer Intensity in Foxp3$^+$ splenic T cells showing a mix of new and arrested transcription states in young mice. This means that some cells initiate Foxp3 transcription with relatively low activity, while others terminate Foxp3 transcription before accumulating substantial Timer proteins. This pattern suggests that new Foxp3 induction in splenic CD4$^+$ T cells in young neonates frequently leads to premature termination. Traditionally, these initiating cells may be viewed as peripherally-induced Tregs, or pTregs[58], and the terminating ones as ex-Tregs[59]. However, such classifications do not capture the nuanced continuous dynamics of Foxp3 transcription our study has revealed. Future research should conduct quantitative, temporal analyses of Foxp3 transcriptional activities using data-driven methods, further refining the ML approaches introduced in this study.

Furthermore, the progressive decline of Foxp3 transcriptional dynamics in splenic T cells with ageing, as illustrated by the TockyConvNet continuous score (Fig. 7c), demonstrates how ageing progressively modifies the temporal dynamics of Foxp3 transcription in peripheral T cells, providing insights into T cell senescence. The arrested Foxp3 transcription in aged splenic T cells, characterised by high Timer Angle and low Timer Intensity (Fig. 9c), indicates that transcriptional activities become infrequent and weak as mice age. Although previous studies described an increased Foxp3$^+$ T cells in the periphery of aged mice[25,60], the biological significance of this increase remains unclear. The progressive increase in the percentage of Foxp3 Timer$^+$ cells (Fig. 6e) alongside the steady decline of Foxp3 transcriptional frequency in its homoeostatic regulation indicates that splenic Foxp3 transcriptional dynamics are shaped by progressive changes induced by ageing, affecting both the quantity and composition of Foxp3$^+$ T cell fraction. These changes are thus likely driven by ageing-related epigenetic modifications such as global shifts in methylation patterns[61] and targeted methylation in key genes[62]. These modifications may directly impact Foxp3 transcription or indirectly influence it through signalling molecules and transcription factors involved in Foxp3 transcription, such as TCR, IL-2, and TGF-β signalling downstream[13].

In summary, our study demonstrates how intact CNS2 controls Foxp3 transcription in the presence of undisturbed Foxp3 protein, addressing a key aspect of Foxp3 autoregulation[9]. It also reveals age- and development-dependent changes in Foxp3 transcription within both the thymus and periphery. These findings are validated by quantitative model performance metrics, ensuring transparency and reproducibility. Our study thus establishes a proof-of-concept for modern data-oriented methodologies that integrate ML with CRISPR and Tocky to unravel dynamic gene regulation at single-cell resolution, which would not be possible by existing methods. This approach overcomes the limitations of traditional immunological research practices, particularly the narrative-driven interpretations of endpoint KO analysis and manual gating. The ambiguity and subjectivity inherent in these traditional practices likely contribute to the ongoing reproducibility crisis in scientific research, especially in immunology and flow cytometric analysis[27,63].

## Methods

### CRISPR-mediated mutagenesis of Foxp3-Tocky

The Foxp3-Tocky mouse strain (BAC Tg (Foxp3$^{\Delta Exon3\ FastFT}$)) was generated by the Ono group and reported previously[7]. The CRISPR-mediated mutagenesis of the Foxp3-Timer transgene in Foxp3-Tocky mice was performed using the previously reported CRISPR transgenic approach[64,65]. Briefly, fertilised eggs were obtained from Foxp3-Tocky and were subjected to electroporation to introduce the Cas9 protein (317-08441, Nippon Gene, Toyama, Japan), tracrRNA (GE-002, FASMAC, Kanagawa, Japan), synthetic crRNAs, and a single-stranded oligodeoxynucleotide (ssODN). The CRISPR strategy was designed to replace the CNS2 sequence with a short insert sequence, AAGTT-TAAAC. Accordingly, the following synthetic crRNAs were used: (1) 5' crRNA targeting CCTGAGCTCCATTATGACAG (PAM: AGG) and (2) 3' crRNA targeting AGTTCCACAAGTATTTAAGG (PAM: AGG). The ssODN designed for homology-directed repair and the insertion of the short nucleotide sequence AAGTTTAAAC/GTTTAAACTTC was 5'-CTCTTTATGTTTGGTCAGAACTTATAAGAAATCTCCTCCT**GTTTAAACTTC**AGAGGATTGGAAAACCCTCTACTGTCCTGATCTGGGGTC-3'. All animal experiments were approved by the Animal Welfare and Ethical Review Body at Imperial College London and the Animal Experiment Committee at Kumamoto University. Mice were housed under a specific pathogen-free (SPF) condition with a 12-h light/dark cycle, at an ambient temperature of 20–24°C and relative humidity of 45–65%. The CNS2 KO Foxp3-Tocky mouse model generated in this study is available from the corresponding author upon reasonable request under a material transfer agreement (MTA).

### ChIP-seq analysis

ChIP-seq data analysis was performed using the NCBI SRA dataset DRP003376[66]. Briefly, HISAT2[67] was used to align sequence reads to the mouse genome (mm10), followed by the data processing using samtools[68] to produce sorted BAM files. The sequence peaks were visualised by Integrative Genomics Viewer[69].

### Timer data preprocessing

Methods for Timer fluorescence data pre-processing, implemented as the R package TockyPrep[41], were reported previously[7]. Briefly, immature blue and mature red fluorescence data are normalised to treat both types of fluorescence equally and thresholded to remove autofluorescence. This is followed by a trigonometric transformation of the normalised data, converting Timer fluorescence into polar coordinates, specifically Timer-Angle and Timer-Intensity.

### TockyKmeansRF implementation

TockyKmeansRF is implemented within the R *TockyRandomForest* package, part of the *TockyMachineLearning* package suite. It integrates k-means clustering[70] and RF classification by *randomForest*[71] utilising the Tocky data preprocessing method *TockyPrep* (v0.1.3)[72]. Briefly, preprocessed training and test datasets undergo *k*-means clustering separately. The default setting for k-means is to use the number of random starts at 1 with the maximum number of iterations set to 10. After k-means clustering, the percentage of cells within each cluster is computed per sample, forming a table of cluster percentages for each dataset.

To match the clusters between the training and testing datasets, first, Timer Angle and Intensity data are standardised. Next, each cluster's representative point is calculated as:

$$\boldsymbol{m}_i = \begin{pmatrix} median(angle_i) \\ median(intensity_i) \end{pmatrix} \tag{1}$$

where i indexes the cluster.

Using the representative points $\mathbf{m_{train}}$ for the training clusters and $\mathbf{m_{test}}$ for the test clusters respectively, the distance matrix **D** is computed using the Euclidean norm:

$$D_{ij} = ||\boldsymbol{m}_{train,i} - \boldsymbol{m}_{test,j}|| \tag{2}$$

where $m_{train, i}$ and $m_{test, j}$ are the representative vectors of the i-th training cluster and the *j*-th testing cluster, respectively.

For the optimal assignment of clusters, we address the linear sum assignment problem (LSAP) using the Hungarian method[73]. This method systematically identifies a permutation of the test set cluster indices that minimises the total Euclidean distance between the corresponding clusters from the training and test datasets. Specifically, the LSAP seeks a permutation matrix P that reorders the columns of the distance matrix such that the sum of the diagonal elements of **DP**, i.e. $\min_{P} trace(DP)$ is minimised, where **D** is the distance matrix, and **P** is a permutation matrix that reorders the columns of **D** to optimally align each test cluster with a training cluster. The *solve_LSAP* function from the R package *clue* (version 0.3.66) is utilised for this purpose[73].

A RF model is then trained using the training data as a cluster percentage table, and the mean decrease Gini (MDG) index of the RF model is used to assess the importance of each cluster. The R package *randomForest* (version 4.7-1.2)[71] is used in the implementation. The test data as a cluster percentage table with the matched clusters is used to predict sample identity, generating model performance metrics. The MDG index is then used to calculate the importance score of individual cells within the test dataset. Subsequently, feature cells are identified using a threshold value and are classified into clusters, where appropriate, using a *density-based spatial clustering of applications with noise* the R package *dbscan* (version 1.2.2)[74].

## Image conversion techniques for TockyConvNet

The image conversion and reversion techniques are implemented as integral components of the TockyConvNetR package (version 0.1.0)[75], part of the *TockyMachineLearning* package suite. Following data preprocessing, the conversion of Timer Angle and Intensity data into image data was performed as follows: First, global minimum and maximum values for the angle and intensity measurements were computed. These extremes were used to establish uniform bin edges across all datasets, typically defining 100 equidistant intervals spanning from the global minimum to the global maximum for each variable. The data were then transformed into pixel-style representations by mapping the flow cytometric measurements into a 2D matrix. Each matrix element represents the count of data points within each bin, effectively converting the flow cytometric data into a $100 \times 100$-pixel image, where each pixel corresponds to a bin count. This method provides a consistent and precise pixelated representation of the flow cytometric data for each sample. To revert image data back to original Timer fluorescence data, parameters defining the pixels, such as global minimum and maximum values along with bin widths, are used.

## ConvNet architectures and model training for TockyConvNet

TensorFlow (version 2.10.0) and Keras (version 2.10.0)[76] were used to construct ConvNet models for TockyConvNet and implemented as the python *TockyConvNetPy* package (version 0.1.0)[77], part of the *TockyMachineLearning* package series. The input layer accepts batch image data of size $100 \times 100$ pixels with a single channel representing cell density, as pre-processed by TockyConvNetR as above.

In our convolutional network architecture, each convolutional operation with a $3 \times 3$ kernel with a ReLU activation is immediately followed by a sigmoid-activated pointwise convolution, which serves as a simple attention block within the network[78]. The primary purpose of this layer is to generate an attention map A:

$$A = \sigma(Conv2D(X, (1, 1)))\qquad(3)$$

where σ represents the sigmoid activation function, and X denotes the input to the attention layer. The resulting attention map A, which has a single channel with spatial dimensions equal to X, is broadcast across

all channels of the feature map X to match its dimensions. The broadcasted attention map A′ is then used for element-wise multiplication with X to modulate the feature response:

$$Y = X \odot A'\qquad(4)$$

where Y is the output feature map after applying the attention, $\odot$ denotes element-wise multiplication, and A′ is the broadcasted version of A. Thus, it applies an element-wise multiplication to the preceding feature map, enhancing or suppressing features based on their spatial importance. A subsequent max pooling layer reduces spatial dimensions, followed by a dropout layer. This configuration, including a convolutional layer, an attention block, max pooling, and dropout, is repeated as indicated.

Finally, the convolutional layers feed into a flatten layer that transitions the data. For classification, this is followed by a fully connected dense layer with ReLU activation and a final dropout at the rate of 0.5 before reaching the softmax output layer. For regression, the flattened data is processed through a sigmoid-activated dense layer to generate a continuous score.

For the TockyConvNet application to Foxp3 Timer Neonatal-to-Ageing Data, the ConvNet model was configured to also receive %total Timer$^+$ cells as numeric data. This inclusion ensures a fair and accurate comparison between TockyConvNet and traditional gating methods, which similarly analyse the percentage of cells within the parent population.

Model training employed the Adam optimiser and used a three-fold cross-validation approach. For each fold, models were compiled using either binary cross-entropy or categorical cross-entropy as the loss function, with accuracy as the evaluation metric. Training was typically conducted over 10 to 12 epochs with batch sizes ranging from 4 to 8, and learning rates set between 0.001 and 0.002. Validation data specific to each fold were used to monitor and assess model performance throughout the training process.

## Grad-CAM

The Grad-CAM methodology[40] with an aggregation method was used to identify the regions of input images that most strongly influence the prediction of ConvNet for a chosen target class. We used the prediction score $y_c$ for class c, which is the logit score prior to the softmax function, and each of available convolutional layers whose feature maps $A_k$ are denoted by k. Using TensorFlow's gradient tape function, we computed the gradient of the class score $y_c$ with respect to the feature map activations:

$$\frac{\partial y_c}{\partial A_{ij}^k}$$

Here, i and j indicate the i-th and the *j*-th pixel within the *k*-th feature map. Next, all the feature matrices were pixel-wise aggregated by global average pooling over i,j. First, the scalar weight $\alpha_c^k$, as a weighted factor for the k-th feature map, was obtained:

$$\alpha_c^k = \frac{1}{z}\sum_i \sum_j \frac{\partial y_c}{\partial A_{ij}^k}\qquad(5)$$

Note that Z is the total number of pixels in the feature map (for instance, 10,000 for the resolution $100 \times 100$). Subsequently, the ReLU was applied to retain the feature contributions that positively influence $y_c$:

$$L_{ij}^{Grad-CAM(c)} = ReLU\left(\sum_k \alpha_c^k A_{ij}^k\right)\qquad(6)$$

Next, the outputs from all the samples ($N_c$ samples) belonging to the class c will be summed and aggregated.

$$\bar{L}_{ij}^{Grad-CAM(c)} = \sum_{n=1}^{N_c} L_{ij}^{Grad-CAM(c,n)} \qquad (7)$$

The heatmap is then normalised to have the range of [0,1]. After generating individual heatmaps for each image in the KO and WT groups, the heatmaps were summed within each group to aggregate activation signals across multiple images. Each aggregated heatmap then underwent Gaussian smoothing to suppress noise and enhance the visibility of predominant activation patterns. These smoothed and aggregated heatmaps were used for the analysis of the ontogeny data. For the analysis of CNS2 KO data, a differential matrix was calculated by subtracting the KO group's Grad-CAM heatmap from the WT group's Grad-CAM heatmap. To implement Grad-CAM, we developed a new gradient model for each convolutional layer by retrieving the layer outputs and utilising the *tf.GradientTape* function in TensorFlow (version 2.10.0)[79] on Jupyter Lab (3.6.3, https://jupyter.org/).

### RNA-seq data analysis
The RNA-seq data analysed in Fig. 5 was previously reported[7] and is deposited to NCBI GEO with the accession number GSE89481. The R package DESeq2 (version 1.44.0)[80] was used to perform Wald tests and *p*-value adjustment with the Benjamini & Hochberg method. In the cross-dataset analysis of flow cytometric Timer data presented in Fig. 5, WT Feature cells were identified as the top 90th percentile cells based on Attention-Conv2 Grad-CAM scores. KO Feature cells were identified as the bottom 25th percentile cells based on Conv2 Grad-CAM scores. Both sets of Grad-CAM scores were derived from the model trained in Fig. 4.

### Logistic regression model and other statistical methods
To quantitatively evaluate the efficacy of manual gating methods, a logistic regression model was developed to predict group classifications based on variables derived from manual gating techniques using the glm function from the R package *stats* (version 4.4.1)[70], specifying a binomial family. The model was trained on a training dataset and validated against an independent test dataset. Predicted probabilities were converted to class labels using a threshold of 0.5. The model's performance was visualised using the R package pROC (version 1.18.5)[81].

Cluster percentage data were analysed by a Kruskal test followed by post-hoc Dunn's test using the R package *dunn.test* (version 1.3.6) with *p*-value adjustment using Bonferroni's correction. Pair-wise comparisons of percentage data were conducted using the Mann-Whitney U test. Mean Fluorescence Intensity (MFI) data were obtained from logarithmically transformed values to achieve normality, and the Student's *t*-test was applied. Where required, *p*-values obtained from these tests were adjusted for multiple comparisons using the Benjamini-Hochberg method.

### Computational performance
The computational performance was assessed using a system with an Apple M2 Ultra processor, 128 GB of memory, and running macOS Sonoma 14.7.1. Analyses were performed using R (version 4.4.1) and Python (version 3.8.15). Runtime was assessed using the *system.time* function in R or the *time.time* function in Python. The memory usage during the execution was monitored using the R package *profmem* (version 0.6.0) and Python's *memory_profiler*.

### Illustrations
All illustrations, including depictions of the mouse model and schematic diagrams, were originally created by the corresponding author using Adobe Illustrator (2025). The figures incorporate the publicly available structural data mCherry chromophore (RCSB PDB ID: 2H5Q[82]) for Fig. 1b. Figure 1a incorporates two Unicode emojis: 'Warning Sign' (U+26A0 U+FE0F) and 'Hand with Fingers Splayed' (U+1F590 U+FE0F) both used in accordance with the Unicode License Version 3[83].

### Reporting summary
Further information on research design is available in the Nature Portfolio Reporting Summary linked to this article.

## Data availability
The data supporting the findings of this study are available at https://github.com/MonoTockyLab/TockyConvNetPy, https://github.com/MonoTockyLab/TockyConvNetR, and https://github.com/MonoTockyLab/TockyRandomForest.

## Code availability
The code used to develop the models, perform the analyses, and generate results in this study is publicly available and has been deposited in GitHub at: https://github.com/MonoTockyLab/TockyConvNetR https://github.com/MonoTockyLab/TockyConvNetPy https://github.com/MonoTockyLab/TockyRandomForest The specific versions of the code associated with this publication are archived in Zenodo and are accessible via the following DOIs: TockyConvNetR (v0.1.0): https://doi.org/10.5281/zenodo.15626073[75] TockyConvNetPy (v0.1.0): https://doi.org/10.5281/zenodo.15626179[77] TockyRandomForest (v0.1.0): https://doi.org/10.5281/zenodo.15626229[45]

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

## Acknowledgements

MO was supported by a CRUK Programme Foundation Award (DCRPGF \100007) and the MRC grant (MR/S000208/1). This research was also supported by KAKENHI research grants from the Japan Society for the Promotion of Science (JSPS) (JP21K07082, JP21H00433, and JP24K10259 to MO), JSPS Core-to-core program to MO and YS (JPJSCCA2020008 to MO and YS), and Japan Agency for Medical Research and Development (24gm1810001s0103 to MO), Advanced Animal Model Support (AdAMS) (JP16H0627601 and JP22H04922 to KA).

## Author contributions

MO conceived the study and all computational and ML methodologies, designed model architectures, wrote computational codes, and performed all computational analysis. NI, YS, KA, and MO designed experiments. MO and KA designed the CRISPR mutagenesis. NT and KA performed transgenesis. NI and MO established the CRISPR mutant colony and performed all flow cytometric experiments. YS, KA, and MO secured funding. MO wrote the manuscript.

## Competing interests

A patent associated with the ML method in this study has been filed (MO). The remaining authors declare no competing interests.
