## [Transparent Peer Review file · Nature Communications]

Machine Learning-Assisted Decoding of Temporal Transcriptional Dynamics via Fluorescent Timer

Corresponding Author: Dr Masahiro Ono

Version 0:

Reviewer comments:

Reviewer #1

(Remarks to the Author)

In the study "Machine Learning-Assisted Decoding of Temporal Transcriptional Dynamics via Fluorescent Timer," Irie et al. use machine learning alongside molecular biology techniques to explore the transcriptional dynamics within regulatory T-cells. Utilizing a genetically modified Foxp3-Tocky mouse model, where CNS2 enhancer region has been removed via CRISPR, the authors aim to observe how CNS2 influences the temporal modulation of Foxp3 transcription.

The authors introduce two ML models to analyze the temporal data: TockyKmeansRF, which integrates k-means clustering with a Random Forest classifier to identify specific cell clusters showing CNS2-dependent regulation, and TockyCNN, a convolutional neural network designed to convert Timer fluorescence data into image formats for classification and feature identification. Together, these models allow for the discovery of "feature cells," whose transcriptional properties are regulated by CNS2, providing insights on this enhancer's role in sustaining Foxp3 expression.

This integrative approach establishes a framework for studying transcriptional dynamics with precision and detail, illustrating the potential of machine learning to deepen biological insights. While this research represents a significant contribution a few points should be addressed to align with the publication standards for Nature Communications. Addressing these points would improve the study's relevance and robustness, positioning it to make a broader impact in the transcriptional regulation field.

1. Machine learning applications are not new in biology, but this study uniquely combines Fluorescent Timer proteins with ML models to explore transcriptional dynamics at the single-cell level, particularly in the context of enhancer function. This enables the analysis of elements like CNS2 in ways that traditional methods may not adequately capture, both dynamically and accurately. Thus, the integration of ML with real-time transcriptional data in this work stands out as innovative.

By focusing on transcriptional dynamics at the single-cell level, the study adds valuable insights into the role of enhancers in gene regulation. Temporal tracking of transcriptional changes at the single-cell level brings forward new perspectives on the role of CNS2 in regulating Foxp3, especially within immune cells, making it a valuable addition to the transcriptional regulation research.

2. Study has clear limitations; particularly with the small sample size (n=10-20). The small sample size leads to higher chance of overfitting, especially in TockyCNN. Expanding the sample size or validating the model on an independent dataset would help the study.

3. The deletion of CNS2 is intended to provide insights into Foxp3 transcriptional regulation over time. Additional validation using well-established temporal markers could strengthen the findings, potentially reinforcing the study's claims about CNS2's role as a regulator.

4. Specific "feature cells" identified by the ML methods have been used to differentiate between WT and knockout groups. Further details on how these cells correlate with broader regulatory mechanisms would improve understanding of their biological significance.

5. TockyCNN employs Grad-CAM to enhance visualization. A sensitivity analysis on how changes in ConvNet architecture

affect Grad-CAM outputs could provide insight into the robustness of these visualizations, thereby supporting confidence in the suggested feature cells.

6. The study normalizes and transforms Timer fluorescence data. A discussion on how these approaches impact data quality and model output, and current challenges and potential solutions would help.

7. Further elaboration on how the new approach outperforms traditional flow cytometry gating techniques in terms of sensitivity and precision would be useful.

8. To facilitate wider use of their code, they should make sure that the GitHub repository include well-documented example datasets and detailed usage instructions for researchers.

9. While the machine learning models are explained, information on the computational resources required for TockyKmeansRF and TockyCNN is absent. Discussing runtime, memory requirements, and scalability for larger datasets would be beneficial for those interested in applying this approach to more complex or large-scale studies.

10. This study primarily investigates the dynamics of Foxp3 in T-cells. It would be helpful to discuss how the toolkit could be adapted for other transcriptional systems or fluorescent Timer-based studies.

11. Iterative k-means clustering is described as monitored for stochastic variation; however, further specifics-such as the number of iterations and stopping criteria-would add clarity.

12. CNS2's role in Foxp3 expression is explored, yet expanding on the implications of these findings would be beneficial.

13. Converting Timer fluorescence data into image-based formats for ConvNet input is a novel approach. However, it would be valuable for the authors to discuss whether changes in bin size or resolution impact classification accuracy.

14. Minor grammatical improvements could improve clarity, such as revising "and and" to "and", and "The Timer fluorescence is expressed in synchronised manner" to "The Timer fluorescence is expressed in a synchronized manner"

(Remarks on code availability)

The code is not made available yet.

Reviewer #2

(Remarks to the Author)

The authors utilized two novel machine learning algorithms to analyze and delineate the difference between WT Foxp3-Tocky cells and CNS2 enhancer deficient Foxp3-Tocky cells in their attempt to identify key features of Foxp3 transcription that is dependent on the CNS2 enhancer. However, the manuscript suffers from the lack of comparative analyses to demonstrate the superiority of the new method compared to more traditional methods, including flow cytometric analyses, and from the lack of a clear conclusion of what the new distinguishing features are. The main problem with the observations is that, at face value, they have limited principal novelty and appear rather confirmatory of previous findings, namely, "that CNS2 may be redundant at the new phase of Foxp3 expression but contributes to the maintenance of Foxp3 expression at the moment when T-cells actively transcribe Foxp3 (persistent locus)". Additional specific concerns are appended below. Considering the multiple issues, enthusiasm for publication of this manuscript in Nature Communication is limited.

1. The overall scope of the analysis and the amount of data in the manuscript is underwhelming as the first three Figures in a "seven-figure: manuscript are all diagrams.

2. Insufficient characterization of either algorithm is provided to demonstrate the superiority of their approach over other methods. The "feature cells" identified to best distinguish WT and CNS2 KO cells can be clearly seen via flow cytometry as shown in Figure 4a. Furthermore, the identified marker changes identified were not validated by flow cytometry.

3. Although some surface marker changes were shown, it is unclear how these changes are dependent on the CNS2 enhancer or what is the biological meaning for the "feature cells" identified. Are these feature cells a distinct subset of Treg cells, or are they in a distinct stage? As the feature cells only make up ~10% of total Treg cells, and the surface marker MFI changes when compared to other timer+ cells are minor (Figure 7J) significance of the observed remains uncertain.

4. Some of the findings are contradictory. For example, the WT feature cells identified in Figure 7, that are most important for distinguishing between WT and KO, had a low score according to clustering shown in Figure 5F.

5. The authors made a point that the CRISPR editing is restricted to BAC transgene but not the endogenous Foxp3 locus. However, no data was shown to support this claim. The PCR experiments were all done with a primer specific to BAC allele, and thus could not characterize the endogenous locus.

Minor concern:

The switching of color schemes in figure 6 is confusing.

(Remarks on code availability)

Reviewer #3

(Remarks to the Author)

(Remarks on code availability)

Reviewer #4

(Remarks to the Author)

In this manuscript entitled “Machine Learning-Assisted Decoding of Temporal Transcriptional Dynamics via Fluorescent Timer”, Irie et al. have elegantly integrated CRISPR with their Tocky system and advanced machine learning methods to investigate the role of the CNS2 enhancer in regulating Foxp3 gene expression in regulatory T (Treg) cells, which I believe represents a seminal contribution to Foxp3+ Treg cell biology, but also to molecular biology in general. Specifically, by using novel machine learning tools (TockyKmeansRF, TockyCNN), the authors have convincingly succeeded in analyzing complex fluorescence profiles of Foxp3-Tocky reporter mice and thus discovered previously unknown functional roles for the CNS2 enhancer, thereby substantially expanding our understanding of Foxp3 gene regulation in Treg cells. Clearly, the integration of CRISPR-mediated gene editing, Tocky technology and two machine learning techniques represents a highly innovative and novel multidisciplinary approach suitable to answer complex biological questions with high precision. Another strength of the study is its robust and innovative design that takes advantage of CRISPR-mediated CNS2 knockout in the established Foxp3-Tocky model to specifically investigate the role of the enhancer without interfering with the endogenous Foxp3 gene locus, overcoming the limitations of previous studies. A major advantage of the newly developed TockyKmeansRF and TockyCNN tools compared to standard methods is the identification of so-called “feature cells”, which represent unique transcriptional dynamics within Treg cells, allowing an unprecedented in-depth understanding of their heterogeneity and gene regulatory mechanisms. In fact, existing methods fail to provide such a nuanced analysis of Foxp3 gene expression dynamics.

In summary, the study substantially enriches the multidisciplinary fields of molecular biology, immunology and machine learning, and I am convinced that the provided novel insights into CNS2's regulatory role in Foxp3 gene transcription obtained through cutting-edge methods are of high value for the field paving the way for future studies. In addition to an important advance in Treg cell biology and molecular immunology, I also expect this work to have an impact on related fields such as molecular biology and transcription regulation. Overall, I strongly support the publication of this manuscript in Nature Communications, as the exceptional quality, originality and significant scientific impact of the study are, in my opinion, perfectly in line with the journal's scope to publish high-quality research across multiple disciplines. Additional minor comments:

- 1) To help clarify the authors may consider highlighting the strengths and unique features of the machine learning approaches (e.g. by implementing a table for comparison, including the trigonometric transformation approach, the two machine learning methods, conventional flow cytometric gating, etc.)
- 2) It would also be useful to include data into the revised manuscript to distinguish between the wild-type and CRISPR knockout Foxp3 Timer BAC (PCR and/or Sanger sequencing?).
- 3) Please explain the details of CRISPR strain selection and establishment during transgenesis of embryos in the revised manuscript. Please clarify: Based on the information in the present manuscript, it remained unclear to me whether endogenous Foxp3 mutants were excluded (e.g. by breeding, and if so, how was this achieved?).
- 4) In my opinion, the discussion part would benefit from a more thorough elaboration of the strengths and broader implications of the newly developed machine learning methods. Ideally, the authors should articulate these points without relying on Tocky-specific jargon (NPt, PAt, Timer Angle, etc.). I believe that improved clarity will make the results more accessible and increase their relevance by linking them more explicitly to the well-established functions of CNS2.
- 5) Please ensure that the authors' computational package is made publicly available, as many Treg biologists and molecular biologists working on gene transcription are likely to be interested in using it.

(Remarks on code availability)

Version 1:

Reviewer comments:

Reviewer #1

(Remarks to the Author)

I believe that the authors have adequately addressed all of my concerns, and I find their revisions satisfactory. The manuscript is now well-prepared for publication, and I fully support its acceptance in Nature Communications.

(Remarks on code availability)

NA

Reviewer #2

(Remarks to the Author)

The authors have performed additional computational analysis to attempt to address some of the concerns raised in the initial review. While it offered clarity on some of the issues (i.e. the CRISPR strategy, and the state of the "feature cells"), the two main issues, 1) the novelty of the discovery and 2) superiority/advantage of the ML method over traditional manual gating of flow cytometry data have not been satisfactorily addressed. The added thymus vs spleen ageing analysis is performed rather crudely and fails to strengthen the manuscript. Thus, the enthusiasm for publishing this manuscript remains limited.

Specific comments to follow:

CNS2 was initially discovered to oppose TCR signaling induced cell-cycle-dependent loss of Foxp3 expressions in an IL-2-STAT5 dependent fashion. Thus, the CNS2 dependent cells (cells that are only present in WT mice) will have higher TCR as well as IL-2 signaling signatures. This is hardly a novel discovery.

The validation of the new methods and its comparison with flow cytometry is based on RNA-seq of FACS-sorted cells and comparisons of marker MFIs of feature cells and "other cells". These validation methods demonstrate that the feature cells can indeed be identified and isolated using flow cytometry. It is unclear whether the "transcriptional dynamics" of CNS2-dependent Foxp3 expression can be discovered by sorting and sequencing Timer+ cells that are only present in WT mice (those that are top left in a timer red vs timer blue plot) vs those only present in KO mice (timer red+ timer blue- cells).

The application of the ML methods to characterize the effect of ageing on transcriptional dynamic of Foxp3 appears rudimentary. As mouse age, the composition of thymic CD4+ T cells changes, the observed differences in timer red/timer blue ratio could be explained by the increased presence of mature Treg cells recirculating from the periphery and consequently not actively upregulating Foxp3. This difference in Foxp3 transcriptional dynamics in CD4 T cells cannot be simply attributed to ageing. Thus, the merit of the analysis shown in Figures 6 and 7 is questionable.

(Remarks on code availability)

Reviewer #3

(Remarks to the Author)

(Remarks on code availability)

Reviewer #4

(Remarks to the Author)

In my opinion, the revised manuscript adequately addresses all of the reviewer's comments and concerns, so I strongly support publication of this very interesting and innovative work.

(Remarks on code availability)

Version 2:

Reviewer comments:

Reviewer #3

(Remarks to the Author)

I co-reviewed this manuscript with one of the reviewers who provided the listed reports. This is part of the Nature Communications initiative to facilitate training in peer review and to provide appropriate recognition for Early Career

Researchers who co-review manuscripts.

(Remarks on code availability)

Point-by-Point Response

Reviewer #1 (Remarks to the Author): In the study “Machine Learning-Assisted Decoding of Temporal Transcriptional Dynamics via Fluorescent Timer,” Irie et al. use machine learning alongside molecular biology techniques to explore the transcriptional dynamics within regulatory T-cells. Utilizing a genetically modified Foxp3-Tocky mouse model, where CNS2 enhancer region has been removed via CRISPR, the authors aim to observe how CNS2 influences the temporal modulation of Foxp3 transcription.

The authors introduce two ML models to analyze the temporal data: TockyKmeansRF, which integrates k-means clustering with a Random Forest classifier to identify specific cell clusters showing CNS2-dependent regulation, and TockyCNN, a convolutional neural network designed to convert Timer fluorescence data into image formats for classification and feature identification. Together, these models allow for the discovery of “feature cells,” whose transcriptional properties are regulated by CNS2, providing insights on this enhancer’s role in sustaining Foxp3 expression.

This integrative approach establishes a framework for studying transcriptional dynamics with precision and detail, illustrating the potential of machine learning to deepen biological insights. While this research represents a significant contribution a few points should be addressed to align with the publication standards for Nature Communications. Addressing these points would improve the study’s relevance and robustness, positioning it to make a broader impact in the transcriptional regulation field.

1. Machine learning applications are not new in biology, but this study uniquely combines Fluorescent Timer proteins with ML models to explore transcriptional dynamics at the single-cell level, particularly in the context of enhancer function. This enables the analysis of elements like CNS2 in ways that traditional methods may not adequately capture, both dynamically and accurately. Thus, the integration of ML with real-time transcriptional data in this work stands out as innovative.

By focusing on transcriptional dynamics at the single-cell level, the study adds valuable insights into the role of enhancers in gene regulation. Temporal tracking of transcriptional changes at the single-cell level brings forward new perspectives on the role of CNS2 in regulating Foxp3, especially within immune cells, making it a valuable addition to the transcriptional regulation research.

Response: We appreciate the reviewer’s insightful analysis of our manuscript and its values. We agree that the temporal tracking transcriptional changes at the single cell-level open new avenues for transcriptional regulation research.

Reviewer #1: 2. Study has clear limitations; particularly with the small sample size (n=10-20). The small sample size leads to higher chance of overfitting, especially in TockyCNN. Expanding the sample size or validating the model on an independent dataset would help the study.

Response:

We greatly appreciate the insightful comments and feedback from Reviewer #1. In response to concerns about the small sample size, we have expanded our datasets significantly. For the CNS2 KO dataset, we increased the sample numbers to 34 and obtained an independent dataset comprising 49 samples. Generating these larger datasets involved the analysis of four times the number of animals from two hemizygote transgenic strains: CNS2 KO Foxp3 Tocky and WT Foxp3 Tocky.

Additionally, we have broadened the scope of our study by including new datasets termed the 'Foxp3 Development-to-Aging Data,' which explore the effects of development and aging on Foxp3 temporal dynamics (**new Figures 6-9**). This includes data from the spleen and thymus of WT Foxp3-Tocky mice across various ages, from neonates to aged mice. The training subset of this data includes 102 samples, with an independent test set of 65 samples.

These enhancements not only allowed for more robust model analysis and optimization but also led to significant immunological discoveries regarding the dynamic nature of Foxp3 transcription across different ages, both in the thymus and the periphery (refer to **new Figure 6**). Furthermore, we optimized the architecture of the TockyConvNet models using the Grad-CAM approach, detailed in **new Figure 8** and **Supplementary Figures S3-S5**. We have also developed two novel TockyConvNet models: a classifier for splenic and thymic Foxp3 transcription dynamics by age and a continuous score model that quantitatively assesses the "thymus-ness" and "spleen-ness" of samples based on the age of the mice (**new Figure 7**).

We believe these efforts have significantly strengthened our manuscript and are grateful for the constructive feedback provided by the reviewer.

Reviewer #1: 3. The deletion of CNS2 is intended to provide insights into Foxp3 transcriptional regulation over time. Additional validation using well-established temporal markers could strengthen the findings, potentially reinforcing the study's claims about CNS2's role as a regulator.

Response: We are grateful for the reviewer's suggestion to enhance our study by further validating the temporal regulatory roles of CNS2 in Foxp3 transcription. Precisely speaking, there are no established temporal markers, but we have delved deeper into our analysis of RNA-seq data from flow-sorted Timer-positive WT Foxp3 Timer cells,

specifically focusing on the temporal dynamics of CNS2-dependent feature cells identified by Grad-CAM analysis. Notably, cells in fraction B2, highly enriched with these features, demonstrated a unique transcriptional signature characterized by elevated Foxp3 and NFAT expression, alongside the repression of other downstream TCR signalling genes (**Figure 5e**). This suggests a temporal regulation of Foxp3 transcription in response to TCR signalling.

In our discussion, we expand on these findings by explaining the intermediate Timer Angle values (40–80°) observed in CNS2-dependent cells (**Figures 4g and 5c**), which indicate that these CNS2-dependent cells have more frequent Foxp3 transcription than the majority of other Foxp3+ cells, as supported by our previous investigations (Bending et al, J Cell Biol, 2018). We propose a model where the transcription frequency of Foxp3 is regulated under TCR-driven conditions, potentially in conjunction with IL-2 signalling (refer to Discussion, page 21 “Biologically, our machine learning models...”– page 23, the first paragraph). This model aligns with the critical roles of Foxp3 in modulating IL-2 receptor expression and TCR responsiveness, pointing to complex autoregulatory loops that involve key proteins and signalling pathways.

We believe that these efforts and planned studies significantly strengthen our manuscript's claims regarding the temporal regulatory role of CNS2 and address your insightful feedback.

Reviewer #1: 4. Specific “feature cells” identified by the ML methods have been used to differentiate between WT and knockout groups. Further details on how these cells correlate with broader regulatory mechanisms would improve understanding of their biological significance.

Response:

Thank you for your constructive feedback on elucidating the biological significance of the 'feature cells' identified by our ML methods. To enhance our analysis, we mapped the Grad-CAM outputs to the original Timer fluorescence data, as illustrated in Figures 4g-4h. This mapping allowed us to identify a significant distinction in the transcriptional profiles of T-cells between WT and CNS2 KO Foxp3 Tocky mice, particularly within the Timer Angle range of 40° to 80°, as described above.

Notably, this range, designated as the Frequency Domain, correlates with higher transcriptional frequencies closer to a 45° Timer Angle, suggesting a heightened Foxp3 transcriptional activity. Combined with our new RNA-seq figure (**Figure 5**), which shows that CNS2-dependent feature cells exhibit the highest levels of Foxp3 expression and a unique activation profile influenced by TCR signalling, it supports the hypothesis that

CNS2 plays a critical role in enhancing the frequency of Foxp3 transcription in response to antigen recognition via TCR signals.

These insights have been incorporated into the extended discussion on the potential regulatory mechanisms facilitated by CNS2, specifically in how it might enhance the transcriptional readiness of T-cells in the immune response. This discussion can be found starting on page 21 of our manuscript, under the section titled 'Biologically, our ML models suggest...!'

Reviewer #1: 5. TockyCNN employs Grad-CAM to enhance visualization. A sensitivity analysis on how changes in ConvNet architecture affect Grad-CAM outputs could provide insight into the robustness of these visualizations, thereby supporting confidence in the suggested feature cells.

Response: We appreciate the reviewer's constructive feedback on the importance of conducting sensitivity analysis to assess how changes in ConvNet architecture might affect Grad-CAM outputs and the robustness of these visualizations. In response to this insightful suggestion, we have undertaken a comprehensive Grad-CAM analysis for each convolutional layer across different ConvNet architectures.

This detailed examination revealed that alterations in ConvNet architecture not only modify the model's performance but also lead to qualitatively different models, as each layer potentially captures distinct aspects of the data relevant to our analysis. For instance, the Grad-CAM visualization of the second-level convolutional layer in a ConvNet with two convolutional layers can vary significantly from that in another ConvNet model with a similar structure (illustrated in **new Supplementary Figures S2 and S3**). This occurs because the decision-making process in a ConvNet is jointly performed across all layers, highlighting the significant impact of ConvNet architecture on feature visualisation and model interpretation.

Moreover, our analysis extended to evaluating the discriminating power of each convolutional layer through ROC and PR curve analyses, which are documented in new **Figures 9a and 9b**. These results underscore that unlike applications with clear visual targets (e.g., image recognition), in flow cytometric applications, different convolutional layers may hold the most class-discriminating information for different target classes. This indicates that ConvNet decisions are indeed an aggregate of contributions across layers, and tuning the architecture inevitably leads to different model behaviours and insights.

Therefore, we have concluded in the flow cytometric application, convolutional layers should be carefully selected for Grad-CAM analysis, and this should be done after optimising ConvNet architecture. These results and discussions are elaborated upon in

the supplementary materials, providing a robust foundation for the confidence in our visualised feature cells.

Reviewer #1: The study normalizes and transforms Timer fluorescence data. A discussion on how these approaches impact data quality and model output, and current challenges and potential solutions would help.

Response:

We appreciate the reviewer's interest in the normalization and transformation processes of Timer fluorescence data. Our investigations into these preprocessing steps have revealed significant insights, particularly regarding the impact of autofluorescent cells on model efficacy. Without normalization, autofluorescent cells significantly distort the Grad-CAM analysis, thereby reducing the effectiveness of the ConvNet approach. These findings have been documented as **new Supplementary Figure S4** and discussed extensively on page 20 of the manuscript, in the paragraph beginning with '*The Tocky machine learning approaches developed...!*

Furthermore, to facilitate broader application and standardization of these preprocessing methods, we have developed them into an independent R package, TockyPrep, available at our GitHub site (<https://github.com/MonoTockyLab/TockyPrep>; <https://monotockylab.github.io/TockyPrep/>). Our methodology and its applications are detailed in a recently published paper on arXiv and in BMC Bioinformatics: Ono, M. *TockyPrep: data preprocessing methods for flow cytometric fluorescent timer analysis. BMC Bioinformatics* 26, 44 (2025). <https://doi.org/10.1186/s12859-025-06058-8>).

Finally, here we clarify that the algorithms and methodologies contained within the TockyPrep package, and those discussed in our arXiv and BMC Bioinformatics publications, are distinct and separate from the methods, data, and algorithms presented in this current manuscript.

Reviewer #1: 7. Further elaboration on how the new approach outperforms traditional flow cytometry gating techniques in terms of sensitivity and precision would be useful.

Response:

We thank Reviewer #1 for the opportunity to further elucidate how our new ConvNet models enhance sensitivity and precision compared to traditional flow cytometric gating techniques.

Our comparative analyses, documented in **new Figure 4e** for the CNS2 KO Foxp3 Tocky dataset and **new Figure 8b-8c** for the Neonatal-to-Ageing WT Foxp3 Tocky dataset, reveal significant performance enhancements with the TockyConvNet models. Specifically, for the CNS2 KO Foxp3 Tocky dataset, we validated the model's efficacy

through ROC and Precision-Recall analyses against traditional manual gating methods, such as Quadrant and Polygon gates. The TockyConvNet model demonstrated superior ability to discriminate between genotypes, achieving almost perfect metrics on an independent test dataset, which significantly outperforms the conventional gating approaches that rely on Quadrant for distinguishing Timer Blue and Red positivity and Polygons for high Blue or Red levels (**Supplementary Figure S1 and Figure 4e**).

For the Neonatal-to-Ageing WT Foxp3 Tocky dataset, we optimized and benchmarked different TockyConvNet architectures—specifically, models with varying numbers of convolutional layers. The Conv3-Layers model emerged as the most effective, consistently outperforming not only the other ConvNet models but also the traditional manual gating techniques in both ROC and Precision-Recall analyses (**Figure 8b-8c**). The traditional quadrant and polygon gates provided satisfactory results for some classes such as Thymus-Young and Spleen-Aged samples; however, they were less effective in accurately classifying more complex classes, demonstrating the enhanced capability of TockyConvNet to manage diverse and challenging dataset characteristics.

These findings collectively demonstrate that our TockyConvNet approach not only improves sensitivity and precision in classifying complex biological samples but also provides a more robust and scalable solution compared to traditional gating techniques. The detailed discussion and supplementary figures elaborating on these points are included in our manuscript to underscore the advancements our methods offer in flow cytometric analysis.

Reviewer #1: 8. To facilitate wider use of their code, they should make sure that the GitHub repository include well-documented example datasets and detailed usage instructions for researchers.

Response:

We greatly appreciate the reviewer's suggestion to enhance the accessibility and usability of our computational tools. We have responded by releasing all related code through the TockyMachineLearning package suite, which comprises three distinct packages tailored to specific aspects of our analysis framework:

<https://monotockylab.github.io/TockyMachineLearning/>

1. **TockyRandomForest:** This package facilitates TockyKmeansRF analysis along with associated statistics and visualizations. It is accessible at:

<https://monotockylab.github.io/TockyRandomForest/>

2. **TockyConvNetR** and **TockyConvNetPy** are both essential components of TockyConvNet analysis:

- **TockyConvNetR** This R package provides methods for image conversion and the inverse mapping of Grad-CAM results back to the original Timer fluorescence data. More details can be found at: <https://monotockylab.github.io/TockyConvNetR/>

- **TockyConvNetPy**: This Python package uses Keras and TensorFlow to construct, train, and test ConvNet models, and to perform Grad-CAM analysis. It is accessible at: <https://monotockylab.github.io/TockyConvNetPy>

Each GitHub repository includes comprehensive vignettes and detailed manuals, along with well-documented example datasets from our CNS2 experiments. Importantly, TockyConvNetPy also provides pre-trained ConvNet models. These resources ensure that researchers have clear, step-by-step instructions on how to conduct the preprocessing and post-Grad-CAM analysis in R, while delegating the core model and Grad-CAM analysis to Python. We are confident that these enhancements will facilitate widespread adoption and effective use of our toolkit.

Reviewer #1: 9. While the machine learning models are explained, information on the computational resources required for TockyKmeansRF and TockyCNN is absent. Discussing runtime, memory requirements, and scalability for larger datasets would be beneficial for those interested in applying this approach to more complex or large-scale studies.

Response:

We appreciate the reviewer's request for details on the computational resources required for our TockyKmeansRF and TockyConvNet models. To address this, we have thoroughly analysed both runtime and memory usage, with findings detailed in **Figure 3h** for TockyKmeansRF and **Supplementary Figure S6** for TockyConvNet. Additionally, the architecture and parameter counts of major ConvNet models are documented in **Supplementary Table 3**.

Runtime and Memory Usage for TockyKmeansRF (Figure 3h): During the execution of TockyKmeansRF on the scaled CNS2 KO dataset, the computational performance metrics were rigorously evaluated. The maximum runtime recorded was approximately 3 seconds per training session, and the peak memory usage did not exceed 30 MB, even as the sample size was increased to 136. Notably, variations in the number of trees within the RF model did not significantly impact either runtime or memory usage, indicating efficient scalability for larger datasets.

Runtime and Memory Usage for TockyConvNet Model Learning (Supplementary Figure S6): For the CNS2 KO Foxp3 Tocky data, processed with our Conv 2-Layer model, the runtime varied between approximately 1 to 6 seconds, and memory usage ranged from about 2 GB, across sample sizes of 52 to 416. When utilizing the Neonate-to-Aging

WT Foxp3 Tocky dataset with the more complex Conv 3-Layers model, we observed an increase in both runtime and memory requirements. Runtime extended from 10 to 70 seconds, and memory usage increased from approximately 1.5 to 3 GB as sample sizes grew. This demonstrates the TockyConvNet model's ability to scale with increasing dataset complexity and size, although it also underscores a corresponding rise in computational demands.

These analyses ensure that future users can gauge the feasibility of applying these models to larger and more complex studies, understanding the computational investment required as they scale up their experiments.

Reviewer #1: 10. This study primarily investigates the dynamics of Foxp3 in T-cells. It would be helpful to discuss how the toolkit could be adapted for other transcriptional systems or fluorescent Timer-based studies.

Response:

Thank you for your insightful suggestion. Our study indeed focuses on the dynamics of Foxp3 in T-cells, yet the methodologies we've developed possess a broad potential for application across various transcriptional systems and fluorescent Timer-based studies. We have included the following points in Discussion, page 20, the paragraph starting with 'The Tocky machine learning approaches developed..':

"The Tocky machine learning approaches developed here are anticipated to be broadly applicable to other Fluorescent Timer datasets, facilitated by the normalisation and data transformation methods provided by TockyPrep⁴². By removing the effects of autofluorescence, these methods enable the effective application of the machine learning methods across various Fluorescent Timer systems (Supplementary Figure S4). However, it is important to acknowledge that adjustments in model architecture and training methods may be necessary to accommodate different data structures, as broadly recognised within machine learning communities⁴³. Additionally, the image conversion techniques employed in our approaches could be effectively adapted for other transcriptional reporter systems, such as EGFP, when combined with an appropriate temporal marker. This adaptation would allow the use of the TockyConvNet algorithms, which can provide a more nuanced understanding of dynamic transcriptional events. Nevertheless, such adaptations would likely require extensive experimentation and optimisation to tailor the approach to specific requirements of individual cases."

11. Iterative k-means clustering is described as monitored for stochastic variation; however, further specifics-such as the number of iterations and stopping criteria-would add clarity.

Response:

We appreciate the reviewer's request for further specifics on the iterative k-means clustering process utilized in our TockyKmeansRF algorithm. Initially, our approach followed the default settings of the k-means function in base R, with the maximum number of iterations set to 10 and the number of random starts at 1 (now included in Materials and Methods, page 25).

To explore the effects of adjusting these parameters, we conducted extensive testing, which included varying the number of iterations (`iter.max`) and the number of random initializations (`nstart`). Our analysis revealed that increasing these values did not substantially enhance the clustering performance.

To provide clarity and substantiate our methodological choices, here we have included additional data for the reviewer's reference, showing that changes in `iter.max` and `nstart` values do not significantly impact the Area Under the Curve (AUC) of our models.

Importantly, while revisiting the TockyKmeansRF approach in response to your comments, we identified a major inefficiency in the previous algorithm, particularly when applied to the Foxp3 Neonatal-to-Ageing Dataset. The original method incorporated iterative cycles of clustering followed by Random Forest analysis, and used Convex Hull gating to identify regions within Timer Blue and Red using a test dataset. This approach, however, proved less effective than anticipated and led us to reconsider the use of Convex Hull gating, which not only reduced the power of the Random Forest analysis but also risked misleading the readers toward traditional gating methods.

In response, we have made significant revisions to the TockyKmeansRF algorithm:

1. We have removed the iterative application of the entire clustering phase from the model training process.
2. We have replaced it with an enhanced cluster matching method that employs the Linear Sum Assignment Problem (LSAP) solving technique using the Hungarian method.

These modifications have greatly improved the robustness of cluster matching between training and testing datasets, ensuring there is no information leakage from test data into training data. They also enhance the overall integrity and reproducibility of the model's performance, which is now clearly demonstrated in the newly added Figure 3.

These methodological details are included Materials and Methods in the revised manuscript, together with the details of the ConvNet and Grad-CAM methods.

12. CNS2's role in Foxp3 expression is explored, yet expanding on the implications of these findings would be beneficial.

Response:

Thank you for highlighting the need to further expand on the implications of our findings regarding CNS2's role in Foxp3 expression. As in our response to the reviewer's other comment, we employed RNA-seq data analysis to explore the role of CNS2 in regulating T cell function. In the revised Discussion, we delve deeper into how our findings enhance the understanding of CNS2's regulatory mechanisms (page 21 "*Biologically, our machine learning models....*" – page 23, first paragraph).

13. Converting Timer fluorescence data into image-based formats for ConvNet input is a novel approach. However, it would be valuable for the authors to discuss whether changes in bin size or resolution impact classification accuracy.

Response:

Thank you for your interest in our novel approach of converting Timer fluorescence data into image-based formats for ConvNet analysis. We agree that understanding the impact of bin size and resolution on classification accuracy is crucial for validating the robustness and applicability of our methodology.

To address this, we have conducted a thorough investigation into the effects of different resolutions on model performance. As detailed in **new Supplementary Figure S5**, we experimented with other resolutions, including 25 x 25 and 400 x 400, and found that a bin size of 100 x 100 pixels indeed yielded the highest classification accuracy. This suggests that a resolution of 100 x 100 strikes the optimal balance by maintaining sufficient detail for accurate feature extraction and ensuring a good density of cells across pixels, which is presumably critical for reducing meaningless variability in the data. This finding and discussion are included in page 17.

14. Minor grammatical improvements could improve clarity, such as revising "and and" to "and", and "The Timer fluorescence is expressed in synchronised manner" to "The Timer fluorescence is expressed in a synchronized manner"

Response: Thank you for your careful reading. We have thoroughly checked the manuscript to improve the clarity and accuracy.

Reviewer #1 (Remarks on code availability):

The code is not made available yet.

Response: The code has been made accessible via our GitHub site as detailed above.

Reviewer #2 (Remarks to the Author):

The authors utilized two novel machine learning algorithms to analyze and delineate the difference between WT Foxp3-Tocky cells and CNS2 enhancer deficient Foxp3-Tocky cells in their attempt to identify key features of Foxp3 transcription that is dependent on the CNS2 enhancer. However, the manuscript suffers from the lack of comparative analyses to demonstrate the superiority of the new method compared to more traditional methods, including flow cytometric analyses, and from the lack of a clear conclusion of what the new distinguishing features are. The main problem with the observations is that, at face value, they have limited principal novelty and appear rather confirmatory of previous findings, namely, “that CNS2 may be redundant at the new phase of Foxp3 expression but contributes to the maintenance of Foxp3 expression at the moment when T-cells actively transcribe Foxp3 (persistent locus)”. Additional specific concerns are appended below. Considering the multiple issues, enthusiasm for publication of this manuscript in Nature Communication is limited.

1. The overall scope of the analysis and the amount of data in the manuscript is underwhelming as the first three Figures in a “seven-figure: manuscript are all diagrams.

Response:

Thank you for your insightful comments. We appreciate the opportunity to further clarify the novel contributions and comparative advantages of our machine learning (ML) models in analysing Foxp3 transcription dynamics dependent on the CNS2 enhancer.

Re. The reviewer’s concern: "Lack of comparative analyses to demonstrate superiority over traditional methods (e.g., flow cytometry)."

We have performed benchmarking analyses and demonstrated the superiority of our novel ML models, including our TockyConvNet (previously designated as TockyCNN) and TockyKmeansRF models, over traditional flow cytometric methods (**new Figures 4e**

and 8b-8c). For the CNS2 KO Foxp3 Tocky dataset, our models, validated through ROC and Precision-Recall analyses, have shown significant advantages over traditional manual gating methods such as Quadrant and Polygon gates, which are predominant existing methods in studies using Fluorescent Timer reporters (for example, Yau et al, JBC, 2020; Eastman et al, Cell Reports, 2020). The TockyConvNet and TockyKmeansRF models achieve almost perfect classification metrics on an independent test dataset, significantly outperforming conventional gating approaches that often fail to discriminate subtle phenotypic variations (**Supplementary Figures S1 and Figure 4e**).

Furthermore, we generated a novel dataset by analysing thymus and spleen from WT Foxp3 Tocky mice at various ages (the Neonatal-to-Ageing WT Foxp3 Tocky dataset). The optimization of different TockyConvNet architectures highlighted the Conv3-Layers model as the most effective (**Figure 8a**). This model consistently outperforms not only other ConvNet configurations but also traditional manual gating techniques in both ROC and Precision-Recall analyses, demonstrating enhanced capability in managing complex dataset characteristics (**Figure 8b-8c**).

The novelty of our methods is not only in the superiority in classification of flow cytometric Timer data but also in the data-oriented identification of CNS2-dependent cells (**Figure 4**) and class-specific features in the Neonatal-to-Ageing dataset (**Figure 9**). This point is further discussed and clarified in our response to your other comments below.

Re. "Observations appear confirmatory of prior work on CNS2's role in Foxp3 maintenance.... The main problem with the observations is that, at face value, they have limited principal novelty and appear rather confirmatory of previous findings)."

Response: While previous studies have established the role of CNS2 in maintaining Foxp3 expression, the methods used for gene deletion in these studies did not allow for an investigation into whether and how CNS2 influences the temporal regulation of Foxp3 transcription. Our work extends these findings by revealing novel temporal dynamics regulated by CNS2.

To further clarify this critical point, we have significantly refined our ML methods, TockyKmeansRF and TockyConvNet (previously designated as TockyCNN), to completely bypass limitations of gating. These enhancements have facilitated a deeper understanding of CNS2's effects at a single-cell level:

1. **TockyKmeansRF:** Our updated TockyKmeansRF model now identifies distinct cell clusters instead of relying on conventional convex hull-gated cells. This method revealed three clusters with unique activation profiles within the WT and CNS2 KO Foxp3 Tocky mice. Notably, CNS2 KO T-cells predominantly shift from Clusters 2 and 3

to Cluster 1, showing reduced Timer Intensity and approaching a Timer Angle of 90°, indicative of arrested transcription (**new Figure 3d-3g**). These findings highlight dynamic transcriptional changes that are not readily captured by objective manual gating techniques.

2. TockyConvNet: By comparing Grad-CAM outputs from WT and CNS2 KO Foxp3 Tocky mice (both carrying an intact endogenous Foxp3 gene), we generated a differential map that pinpoints single-cell regions where CNS2 regulates Foxp3 transcription (**new Figure 4g-4h**). This analysis showed that CNS2-dependent cells at the single cell resolution are characterized by intermediate Timer Angle values (40–80°) and high Timer Intensity. This pattern suggests a CNS2-mediated enhancement in the frequency of Foxp3 transcription, particularly under specific signalling conditions. Our analyses reveal that these cells accumulate Foxp3 transcripts and proteins over time, indicative of dynamic transcriptional regulation within the Frequency Domain (45°–90°, Bending et al, J Cell Biol, 2018). The fine-grained insights provided by Timer-based Grad-CAM analysis surpass those achievable through conventional gating approaches, which rely on predetermined thresholds and assumptions.

3. Integration with RNA-Seq Data: Leveraging Grad-CAM outputs and TockyPrep normalization, we mapped these CNS2-dependent features onto flow cytometric data coupled with RNA-seq (**new Figure 5**). This revealed a specific T-cell fraction (the “B2” cluster) enriched in CNS2-dependent cells, characterized by elevated NFAT and IL-2 receptor expression alongside significantly upregulated Foxp3 transcription. Flow cytometry further substantiated these results, showing increased PD-1 and CD25 expression—markers indicative of TCR/IL-2 signalling activity.

Overall, these data support a model in which CNS2, under the influence of TCR and potentially IL-2 signalling, regulates the frequency of Foxp3 transcription. Since CNS2 is bound and controlled by Foxp3 protein itself, the underlying mechanisms likely involve an autoregulatory loop. Moreover, CNS2 is influenced by downstream effectors of IL-2 signalling (e.g., Stat5) and TCR signalling (e.g., CREB and Ets-1). Our revised model proposes that the quality and quantity of TCR and/or IL-2 signals, along with Foxp3 protein, modulate CNS2 enhancer activities to finely tune Foxp3 transcription dynamics. This conceptual advance reveals that CNS2 does more than simply maintain Foxp3 expression; it actively coordinates temporally distinct patterns of transcription, controlling the frequency of Foxp3 transcription in T-cells under the influence of TCR and other signals including IL-2.

2. Insufficient characterization of either algorithm is provided to demonstrate the superiority of their approach over other methods. The “feature cells” identified to best distinguish WT and CNS2 KO cells can be clearly seen via flow cytometry as shown in

Figure 4a. Furthermore, the identified marker changes identified were not validated by flow cytometry.

Response: Thank you for your feedback regarding the characterization of our algorithms and the validation of the distinguishing features identified in our study. We have taken your concerns into account and made substantial improvements to the manuscript to better demonstrate the advantages of our machine learning approaches over traditional methods:

1. We have conducted extensive benchmarking analyses employing performance metrics such as Receiver Operating Characteristic (ROC) curves and Precision-Recall curves (**new Figures 4e and 8**). These analyses demonstrate that our machine learning approaches significantly surpass conventional manual gating methods in capturing CNS2-dependent features. Furthermore, TockyConvNet exhibits exceptional performance on the newly generated Neonatal-to-Ageing dataset, validating its effectiveness across diverse biological conditions.
2. The Grad-CAM method has been significantly enhanced to perform single-cell level analysis. By integrating a reverse mapping technique, we now map Grad-CAM outputs back to the original Timer fluorescence space, which allows for precise analysis of the nuanced dynamics of Timer profiles associated with CNS2-dependent features. This advanced methodology is detailed in new **Figures 4g through 4j**, showing CNS2-dependent cells in a resolution previously unachievable.
3. The 'Materials and Methods' section has been expanded to include detailed mathematical descriptions and methodological frameworks for TockyKmeansRF implementation, ConvNet architectures, model training for TockyConvNet, and the application of Grad-CAM. These additions provide a thorough understanding of how each model operates and the principles behind their analyses.
4. The algorithm of the upgraded TockyKmeansRF is visually elaborated in **Figure 3a**. Precisely, the distance-base matching method enables effective identification of matched clusters in test data.
5. Regarding the validation of flow cytometric data in Figure 4, we have included two independent test datasets. The test data included in the manuscript combines the two datasets and statistical analysis was performed on the combined data. We have included individual analyses of the two datasets in the following two pages:

Test Dataset-1 (in relation to Figure 4k)

P-Values: Test Data 1				
	CD25	CD44	CD69	PD1
Conv1	0.3645	0.0001*	0.053	0.0323
Attention1-Conv	0.0033*	0.0000*	0.0065*	0.102
Conv2	0.047	0.0188*	0.102	0.0006*
Attention2-Conv	0.0921	0.0000*	0.1363	0.0163*

Test Dataset-2 (in relation to Figure 4k)

P-Values: Test Data 2				
	CD25	CD44	CD69	PD1
Conv1	0.3812	0.0001*	0.0247*	0.2984
Attention1-Conv	0.0063*	0.0002*	0.0117*	0.145
Conv2	0.0562	0.0012*	0.0562	0.0653
Attention2-Conv	0.0348	0.0001*	0.0063*	0.2028

6. Moreover, to further substantiate the phenotypic distinctions identified by our machine learning models, we have included a new section on RNA-seq data (presented in **new Figure 5**). This section demonstrates that the flow-sorted fraction of T-cells, which is highly enriched in CNS2-dependent cells, exhibits distinct gene expression profiles. Notably, these cells show significant upregulation of Foxp3 and genes associated with TCR signalling and IL-2 pathways. This supports the biological relevance and robustness of our computational findings, providing a strong validation of the phenotypic changes identified through our advanced ML techniques.
7. We have significantly expanded the Discussion section to articulate how our data-driven framework overcomes limitations and potential biases inherent in manual gating. This includes a detailed comparison of traditional gating methods with our machine learning approaches, summarized in **Supplementary Table 4**. This table and discussion highlight the quantitative and objective nature of our analyses, contrasting them with the more subjective and less adaptable nature of manual gating.

3. Although some surface marker changes were shown, it is unclear how these changes are dependent on the CNS2 enhancer or what is the biological meaning for the “feature cells” identified. Are these feature cells a distinct subset of Treg cells, or are they in a distinct stage? As the feature cells only make up ~10% of total Treg cells, and the surface marker MFI changes when compared to other timer+ cells are minor (Figure 7J) significance of the observed remains uncertain.

Response:

The fact that CNS2-dependent cells represent approximately 10% of total Foxp3-expressing cells is partly a reflection of our quantitative analysis approach. To capture the most characteristic phenotype of cells dependent on CNS2, we deliberately focused on the top 10% of cells in the Grad-CAM output. Although the range of cells with WT-like features is broader than 10%, our analysis indicates that these cells are in a transient, dynamic state rather than reflecting a stable, distinct subset. This transient status does not diminish their biological importance; rather, it underscores a critical phase during which Foxp3 expression and regulatory activities are modulated in response to specific signalling cues. The following evidence supports this conclusion:

1. Grad-CAM Analysis Across Convolutional Layers: Our Grad-CAM analysis, performed across multiple convolutional layers, reveals nuanced changes in surface marker expression. For clarify, all MFI values in figures are log-transformed. For example, CNS2-dependent cells (feature cells) exhibit significantly higher marker expression compared to other cells in the last convolutional layer:

CD44: log₁₀ MFI of 4.05 (feature cells) vs. 3.66 (others) [raw MFI: 11,171 vs. 4,528]

CD69: log₁₀ MFI of 2.69 vs. 2.49 [raw MFI: 493 vs. 307]

PD1: log₁₀ MFI of 2.47 vs. 2.38 [raw MFI: 295 vs. 241]

These results are from the pooled and combined data from two independent experiments, as clarified above (Supplementary Table 1).

2. RNA-seq Data (**Figure 5**): Our RNA-seq analysis of CNS2-dependent cells reveals significant upregulation of genes associated with acute T-cell responses, including NFAT and IL-2 receptor components, in addition to highly upregulated Foxp3 itself. This unique gene expression profile supports the notion that these cells engage in transient regulatory functions crucial during specific phases of immune activation.
3. The intermediate Timer Angles and unique changes of gene expression downstream of key signalling pathways indicate that CNS2-dependent cells are not a fixed, stable population but rather represent a dynamic transcriptional state. This state likely reflects a critical phase in which Foxp3 transcription is finely tuned in response to TCR and IL-2 signalling.

Collectively, these findings support that the Foxp3 Tocky system and our ConvNet/Grad-CAM approach have identified a unique, transient state of Foxp3-expressing cells that is driven by CNS2. Our revised Discussion further elaborates on these points and emphasizes the importance of focusing on molecular mechanisms at the single-cell level rather than on static population-level analyses (Discussion, pages 21 – 23).

4. Some of the findings are contradictory. For example, the WT feature cells identified in Figure 7, that are most important for distinguishing between WT and KO, had a low score according to clustering shown in Figure 5F.

Response:

We appreciate the reviewer's observation regarding the apparent discrepancy between the WT feature cells identified by the two machine learning methods. The revised manuscript now includes improved results from the two models, which are more similar to each other than the previous results. Importantly the difference is largely attributable to methodological differences between our two approaches. TockyKmeansRF utilises k-means clustering to identify cell clusters, providing information at a cluster level, whereas TockyConvNet—combined with Grad-CAM analysis—resolves features at the single-cell level.

In our revised manuscript, the first paragraph of the Discussion section clarifies this distinction.: “Notably, the major features identified by the TockyKmeansRF as T-cell clusters (Figures 3d–3f) are resolved at the single-cell level in TockyConvNet. Here, Clusters 2 and 3 identified by TockyKmeansRF, which predominated in WT samples, are captured as CNS2 WT features in the Grad-CAM outputs, while Cluster 1, more prevalent in KO samples, is distinguished as the KO feature primarily within the first three convolutional layers (Figures 4g–4j).”

5. The authors made a point that the CRISPR editing is restricted to BAC transgene but not the endogenous Foxp3 locus. However, no data was shown to support this claim. The PCR experiments were all done with a primer specific to BAC allele, and thus could not characterize the endogenous locus.

Response:

We appreciate the reviewer's request for clarification regarding the specificity of CRISPR editing to the BAC transgene rather than the endogenous Foxp3 locus. To address this concern, we implemented specific experimental strategies to ensure and confirm that the current CNS2 KO Foxp3 Tocky does not carry any mutations in their endogenous Foxp3 gene and the CRISPR-Cas9 editing was restricted to the BAC transgene.

1. CRISPR Strategy: although the CRISPR system itself is not inherently specific to the BAC transgene, we utilized stringent PCR-based screening to isolate mutants where only the BAC transgene was altered.

- Common CNS2 Deletion PCR: Designed to detect potential deletions in both the endogenous and BAC Foxp3-Timer loci.
- Foxp3 Timer-Specific Discrimination PCR: Specifically discriminates between the wild-type Foxp3 Timer and CNS2 KO Foxp3 Timer alleles.

2. Founder Selection and Sequencing: We identified founder mouse #87 using the above PCR assays, confirming that this mouse carried the CNS2 KO mutation exclusively in the Foxp3 Timer transgene with no detectable modifications in the endogenous Foxp3 gene (**Figures 2e–2f**). Sanger sequencing further validated the targeted deletion and precise homologous recombination at the CNS2 locus within the BAC transgene (**Figure 2g**).

3. Strategic Breeding: Over a period of two years, we established a breeding line from founder #87, repeatedly backcrossing to the C57BL/6 background to confirm stable transmission of the CNS2 deletion confined to the BAC transgene (**Figure 2h**). This process robustly excluded any alterations in the endogenous Foxp3 gene.

4. Current Colony Status and Additional Sequencing: The CNS2 KO Foxp3 Tocky transgene is consistently inherited according to Mendelian patterns, indicating stable genetic transmission. The specificity of the Foxp3 Timer-Specific Discrimination PCR, crucial for distinguishing the edited from the unedited alleles, has been further validated by Sanger sequencing (**Figures 2i-2j**)."

These methodological details and results are now included in Figure 2 and the corresponding text part in Results, providing substantial evidence that the CNS2 KO Foxp3 Tocky mice do not carry any mutations in the endogenous Foxp3 locus. All experimental animals used in subsequent studies were derived from this rigorously validated breeding line.

We can also confirm that all CNS2 experiments were conducted using littermates from WT Foxp3 Tocky and CNS2 KO Foxp3 Tocky, excluding double transgenics, as clarified in the revised manuscript.

Minor concern:

The switching of color schemes in figure 6 is confusing.

Response: The figures have been fully updated.

Reviewer #3 (Remarks to the Author):

Reviewer #4 (Remarks to the Author):

In this manuscript entitled "Machine Learning-Assisted Decoding of Temporal Transcriptional Dynamics via Fluorescent Timer", Irie et al. have elegantly integrated CRISPR with their Tocky system and advanced machine learning methods to investigate the role of the CNS2 enhancer in regulating Foxp3 gene expression in regulatory T (Treg) cells, which I believe represents a seminal contribution to Foxp3+ Treg cell biology, but also to molecular biology in general. Specifically, by using novel machine learning tools

(TockyKmeansRF, TockyCNN), the authors have convincingly succeeded in analyzing complex fluorescence profiles of Foxp3-Tocky reporter mice and thus discovered previously unknown functional roles for the CNS2 enhancer, thereby substantially expanding our understanding of Foxp3 gene regulation in Treg cells. Clearly, the integration of CRISPR-mediated gene editing, Tocky technology and two machine learning techniques represents a highly innovative and novel multidisciplinary approach suitable to answer complex biological questions with high precision. Another strength of the study is its robust and innovative design that takes advantage of CRISPR-mediated CNS2 knockout in the established Foxp3-Tocky model to specifically investigate the role of the enhancer without interfering with the endogenous Foxp3 gene locus, overcoming the limitations of previous studies. A major advantage of the newly developed TockyKmeansRF and TockyCNN tools compared to standard methods is the identification of so-called “feature cells”, which represent unique transcriptional dynamics within Treg cells, allowing an unprecedented in-depth understanding of their heterogeneity and gene regulatory mechanisms. In fact, existing methods fail to provide such a nuanced analysis of Foxp3 gene expression dynamics.

In summary, the study substantially enriches the multidisciplinary fields of molecular biology, immunology and machine learning, and I am convinced that the provided novel insights into CNS2’s regulatory role in Foxp3 gene transcription obtained through cutting-edge methods are of high value for the field paving the way for future studies. In addition to an important advance in Treg cell biology and molecular immunology, I also expect this work to have an impact on related fields such as molecular biology and transcription regulation. Overall, I strongly support the publication of this manuscript in Nature Communications, as the exceptional quality, originality and significant scientific impact of the study are, in my opinion, perfectly in line with the journal's scope to publish high-quality research across multiple disciplines. Additional minor comments:

1) To help clarify the authors may consider highlighting the strengths and unique features of the machine learning approaches (e.g. by implementing a table for comparison, including the trigonometric transformation approach, the two machine learning methods, conventional flow cytometric gating, etc.)

Response:

Thank you for your encouraging and comprehensive review of our work. We are pleased that you recognize the novel contributions of integrating CRISPR technology, Tocky technology, and advanced machine learning tools in decoding the regulatory roles of the CNS2 enhancer in Foxp3 gene regulation. Your positive feedback reinforces our belief that this multidisciplinary approach not only pushes the frontiers in the study of regulatory T cells but also extends broadly to molecular biology and immunology.

We have taken your suggestion to more clearly delineate the strengths and unique features of the machine learning approaches used in our study compared to traditional methods. In response, we have developed and included a new Supplementary Table (**Supplementary Table 4**) in the revised manuscript. This table and the first paragraph of Discussion provide a detailed comparison between the innovative machine learning methods utilized (TockyKmeansRF and TockyConvNet) and conventional flow cytometric gating. This table delineates the core methods, strengths, and weaknesses of each approach,

1. Manual Gating (Quadrant and Polygon):

- Core Method: Uses fixed thresholds for marker positivity or arbitrary gating based on 2D plots.
- Strengths: Suitable and accessible for small sample sizes, facilitating use by those inexperienced in complex data analysis.
- Weaknesses: These methods are rigid, limited to predefined cell populations, and do not adapt to new data insights. They also suffer from low resolution and potential reproducibility issues, making them less suitable for nuanced analysis of complex datasets.

2. TockyKmeansRF:

- Core Method: Combines TockyPrep, K-means clustering, and Random Forest analysis.
- Strengths: This approach is dynamic and evolves in response to the data, providing more flexibility than static manual gating.
- Weaknesses: While it evolves to adapt to data, it only outputs cluster-level data, lacking the resolution to provide single-cell insights.

3. TockyConvNet and Grad-CAM:

- Core Method: Utilizes TockyPrep, image conversion, ConvNet, and Grad-CAM for deep analysis.
- Strengths: Facilitates a high-resolution, visually intuitive, and data-driven analysis, offering single-cell resolution that captures dynamic transcriptional changes.
- Weaknesses: There's a risk of overfitting, which necessitates careful calibration of model architecture and training strategies. This method also demands high-quality, substantial datasets for optimal performance.

We believe this additional information will enhance readers' appreciation of the innovative aspects of our approach and the significant advantages it offers over existing methodologies.

2) It would also be useful to include data into the revised manuscript to distinguish between the wild-type and CRISPR knockout Foxp3 Timer BAC (PCR and/or Sanger sequencing?).

Response:

Thank you for your suggestion to include data that clearly distinguishes between the wild-type and CRISPR knockout Foxp3 Timer BAC. In response to your feedback, we have now incorporated detailed PCR and Sanger sequencing results into the revised manuscript (**Figure 2e – 2j**), which provide definitive evidence of the specific genomic modifications achieved by our CRISPR/Cas9 editing.

- 1. PCR Analysis:** We have added results from a PCR assay designed specifically to amplify regions surrounding the CNS2 locus within the BAC transgene. This assay differentiates between the wild-type and knockout alleles by size discrimination, where the knockout allele exhibits a distinct band size due to the deletion introduced by CRISPR editing. These PCR results are now presented in new **Figure 2e, 2f, and 2h**, illustrating clear separation between wild-type and knockout samples.
- 2. Sanger Sequencing:** To further confirm the specific deletions and to ensure the precision of our CRISPR edits, we performed Sanger sequencing on PCR products from both wild-type and knockout samples (**Figure 2g and 2j**). The sequencing data confirm the exact boundaries of the deletion and are consistent with the intended modifications of the CNS2 region. These sequencing traces have been included in new Figure Y, providing a direct visual confirmation of the CRISPR-induced changes at the molecular level.

These additions ensure that our manuscript now includes comprehensive molecular data supporting the specific differentiation between wild-type and knockout Foxp3 Timer BAC alleles, thereby addressing the concerns about the specificity and accuracy of our genetic modifications. We believe that this detailed genetic characterization strengthens the findings and conclusions of our study, offering clear insights into the effects of CNS2 deletion within the BAC transgene context.

3) Please explain the details of CRISPR strain selection and establishment during transgenesis of embryos in the revised manuscript. Please clarify: Based on the information in the present manuscript, it remained unclear to me whether endogenous Foxp3 mutants were excluded (e.g. by breeding, and if so, how was this achieved?).

Response:

We appreciate the reviewer's request for clarification regarding the specificity of CRISPR editing to the BAC transgene rather than the endogenous Foxp3 locus. To address this

concern, we implemented specific experimental strategies to ensure and confirm that the current CNS2 KO Foxp3 Tocky does not carry any mutations in their endogenous Foxp3 gene and the CRISPR-Cas9 editing was restricted to the BAC transgene.

1. CRISPR Strategy: although the CRISPR system itself is not inherently specific to the BAC transgene, we utilized stringent PCR-based screening to isolate mutants where only the BAC transgene was altered.

- Common CNS2 Deletion PCR: Designed to detect potential deletions in both the endogenous and BAC Foxp3-Timer loci.
- Foxp3 Timer-Specific Discrimination PCR: Specifically discriminates between the wild-type Foxp3 Timer and CNS2 KO Foxp3 Timer alleles.

2. Founder Selection and Sequencing: We identified founder mouse #87 using the above PCR assays, confirming that this mouse carried the CNS2 KO mutation exclusively in the Foxp3 Timer transgene with no detectable modifications in the endogenous Foxp3 gene (**Figures 2e-2f**). Sanger sequencing further validated the targeted deletion and precise homologous recombination at the CNS2 locus within the BAC transgene (**Figure 2g**).

3. Strategic Breeding: Over a period of two years, we established a breeding line from founder #87, repeatedly backcrossing to the C57BL/6 background to confirm stable transmission of the CNS2 deletion confined to the BAC transgene (**Figure 2h**). This process robustly excluded any alterations in the endogenous Foxp3 gene.

4. Current Colony Status and Additional Sequencing: The CNS2 KO Foxp3 Tocky transgene is consistently inherited according to Mendelian patterns, indicating stable genetic transmission. The specificity of the Foxp3 Timer-Specific Discrimination PCR, crucial for distinguishing the edited from the unedited alleles, has been further validated by Sanger sequencing (**Figures 2i-2j**)."

These methodological details and results are now included in Figure 2 and the corresponding text part in Results, providing substantial evidence that the CNS2 KO Foxp3 Tocky mice do not carry any mutations in the endogenous Foxp3 locus. All experimental animals used in subsequent studies were derived from this rigorously validated breeding line.

We can also confirm that all CNS2 experiments were conducted using littermates from WT Foxp3 Tocky and CNS2 KO Foxp3 Tocky, excluding double transgenics, as clarified in the revised manuscript.

4) In my opinion, the discussion part would benefit from a more thorough elaboration of the strengths and broader implications of the newly developed machine learning

methods. Ideally, the authors should articulate these points without relying on Tocky-specific jargon (NPt, PAt, Timer Angle, etc.). I believe that improved clarity will make the results more accessible and increase their relevance by linking them more explicitly to the well-established functions of CNS2.

Response:

Thank you for your constructive feedback regarding the discussion section of our manuscript. We agree that elucidating the strengths and broader implications of the machine learning methods developed in this study, while minimizing the use of Tocky-specific jargon, will enhance the clarity and accessibility of our findings.

In response to your suggestions, we have revised the discussion to more clearly articulate the innovative aspects and potential applications of our machine learning approaches, particularly in the context of transcriptional regulation studies. We have made the following specific changes to improve clarity and broaden the appeal of the discussion:

1. We have carefully reviewed the discussion to remove or define specialized terms such as "NPt" and "PAt". However, 'Timer-Angle' is an essential term and needed to remain. Instead, we included improved introduction to the concept of Timer Angle in Introduction (Result, page 6, *'To use these models, flow cytometric Timer Blue and Red fluorescence data are pre-processed and transformed into Timer Angle and Timer Intensity...'*).
2. We have included new biological discussion paragraphs to clarify the significance of our findings regarding CNS2 and the new discovery of Foxp3 dynamics in developmental and ageing stages.
3. To increase the relevance of our results, we have emphasized how these machine learning tools can be applied to broader research questions in immunology and other fields. We discuss potential applications and how other researchers might use these tools to explore similar questions in different biological contexts.

We believe these revisions will make the results more accessible and increase their relevance by clearly linking them to established immunological functions and potential clinical implications. We hope that these changes will satisfy your concerns and make the discussion both more informative and engaging for a broader audience.

5) Please ensure that the authors' computational package is made publicly available, as many Treg biologists and molecular biologists working on gene transcription are likely to be interested in using it.

Response:

We greatly appreciate the reviewer's suggestion to enhance the accessibility and usability of our computational tools. We have responded by releasing all related code through the TockyMachineLearning package suite, which comprises three distinct packages tailored to specific aspects of our analysis framework:

<https://monotockylab.github.io/TockyMachineLearning/>

1. **TockyRandomForest**: This package facilitates TockyKmeansRF analysis along with associated statistics and visualizations. It is accessible at:

<https://monotockylab.github.io/TockyRandomForest/>

2. **TockyConvNetR** and **TockyConvNetPy** are both essential components of TockyConvNet analysis:

- **TockyConvNetR** This R package provides methods for image conversion and the inverse mapping of Grad-CAM results back to the original Timer fluorescence data.

More details can be found at: <https://monotockylab.github.io/TockyConvNetR/>

- **TockyConvNetPy**: This Python package uses Keras and TensorFlow to construct, train, and test ConvNet models, and to perform Grad-CAM analysis. It is accessible at:

<https://monotockylab.github.io/TockyConvNetPy>

Each GitHub repository includes comprehensive vignettes and detailed manuals, along with well-documented example datasets from our CNS2 experiments. Notably, TockyConvNetPy also provides pre-trained ConvNet models. These resources ensure that researchers have clear, step-by-step instructions on how to conduct the preprocessing and post-Grad-CAM analysis in R, while delegating the core model and Grad-CAM analysis to Python. We are confident that these enhancements will facilitate widespread adoption and effective use of our toolkit.

Point-By-Point Response to Reviewer #2

Reviewer's Comments (Major Points): *'The authors have performed additional computational analysis to attempt to address some of the concerns raised in the initial review. While it offered clarity on some of the issues (i.e. the CRISPR strategy, and the state of the "feature cells"), the two main issues, 1) the novelty of the discovery and 2) superiority/advantage of the ML method over traditional manual gating of flow cytometry data have not been satisfactorily addressed. The added thymus vs spleen ageing analysis is performed rather crudely and fails to strengthen the manuscript. Thus, the enthusiasm for publishing this manuscript remains limited.'*

Response: We appreciate your acknowledgment of our clarifications regarding the CRISPR strategy and feature cell characterization. However, we respectfully disagree with your assessment regarding the novelty and methodological advantages of our work. Below, we address each point, emphasizing how our discoveries regarding physiological dynamics of T cells were uniquely enabled by our integrated Tocky-CRISPR-ML approach, surpassing what is achievable by existing KO methods or manual gating.

A. Novelty of Discovery (CNS2): Our study uniquely demonstrates at **single-cell resolution** how CNS2 dynamically drives **high-frequency** Foxp3 transcription bursts. This finding challenges the prevailing view that CNS2 merely 'maintains' Foxp3 expression, a perspective derived from endpoint, bulk analyses of KO strains that fail to distinguish between primary effects of deletion from secondary effects due to attenuated Foxp3 protein levels. Our approach explicitly addresses this crucial distinction, which is important given the autoregulatory loop of Foxp3 gene regulation mediated by Foxp3 protein itself (see 1.1 – 1.3 below).

B. Advantages of the ML Method Against Manual Gating: We reiterate that our ML methods surpass manual gating, as established through extensive benchmarking and **quantitative** model-performance analysis, consistently demonstrating **superior** performance of TockyConvNet, compared with manual gating. Two independent experiments (the CNS2 and the **neonatal-to-ageing** datasets) **robustly** validate the performance and generalizability of our ML approach (**Figs. 4e, 4f, and 8**, p. 24 line 12–16).

In response to your additional request for clarity, we have further clarified the aim of our study (p. 5 line 14–19) and included **new Figs. 1a and 1b**, which illustrate the inherent limitations of manual gating and the clear advantages offered by our ML approach, presenting our research framework, showing how ML model training and model-behavior analysis identify **group-specific features** based solely on the pattern of Timer data. In this framework, the rigor of new biological discoveries is supported by **quantitative** model performance metrics, achieving a level of rigor and transparency unattainable by manual gating (p. 6 line 20–p. 8 line 4).

C. The Ageing Dataset and ML Framework Are Rigorous, Not "Crude": We respectfully disagree with your characterization of our ageing dataset and analysis as "crude." Rather, our approach provides an unprecedented quantitative standard for examining Foxp3 transcriptional dynamics across the lifespan in two immunologically critical organs (spleen and thymus). Our inclusion of four distinct age and tissue categories, with clear rationales for young versus aged classification, is supported by robust scientific justification (p. 19, line 18–p. 20, line 8). Specifically, the dataset and analysis achieve the following three points:

1. Biological Novelty: Prior studies have focused exclusively on thymic neonatal stages¹ or compared only limited timepoints (adult vs. aged)^{2,3}, leaving critical gaps in understanding age-dependent processes (p. 5 line

1–10). Our comprehensive dataset uniquely bridges this gap, spanning the full lifespan in both spleen and thymus. Leveraging this dataset, TockyConvNet has uncovered previously unknown physiological Foxp3 transcription dynamics influenced by developmental and ageing-dependent processes, not only in the thymus but also in the spleen (**Figs. 7 and 9**), making significant advancements (Abstract, p. 28 line 7–p. 30 line 15).

2. Technological Rigor and Innovation: The ageing dataset was carefully generated and optimized explicitly for benchmarking our ML methods, achieving high-quality standards critical for ConvNet training (**Figs. 4e, 4f, 8**). Our four-class classifier (young vs. aged, thymus vs. spleen) integrates all classes into a single model, achieving **excellent** performance metrics that surpass manual gating and enabling interpretable feature identification via Grad-CAM (**Figs. 8, 9**). These technological advances significantly enhance transparency, quantitative rigor, and reproducibility—achievements not possible with traditional gating or preceding immunological methods (p. 30 line 16–).

3. Open Data and Reproducibility: Finally, all datasets, ML models, and computational codes have been made publicly accessible (p. 39, lines 22–23). This transparency enables unprecedented reproducibility and independent validation, exceeding the standards established by previous studies.

1. Reviewer’s Comments: *“CNS2 was initially discovered to oppose TCR signalling induced cell-cycle-dependent loss of Foxp3 expressions in an IL-2-STAT5 dependent fashion. Thus, the CNS2 dependent cells (cells that are only present in WT mice) will have higher TCR as well as IL-2 signalling signatures. This is hardly a novel discovery.”*

Response: Thank you for your feedback. We clarify that our findings expand **significantly** beyond the role of CNS2 in maintaining Foxp3 expression, as demonstrated by earlier studies, which were unable to address in which specific Foxp3⁺ cells intact CNS2 functions. Using the unique advantages of our integrated CRISPR system—where the endogenous Foxp3 remains intact—we present compelling evidence that CNS2 induces **high-frequency** Foxp3 transcriptional dynamics at the **single-cell level**, effectively addressing the autoregulatory loop of the Foxp3 gene involving Foxp3 protein itself⁴. Thus, our findings **challenge** the assumptions from bulk analysis of complete KO endpoints as follows:

1.1. What Previous Studies Found for CNS2 regarding Foxp3 expression under TCR/IL-2 signalling:

Previous foundational CNS2 studies by the Rudensky and Zheng groups demonstrated the following evidence regarding Foxp3 and TCR/IL-2 signalling dynamics (p. 4, line 4–24; p. 27 line 19–20):

- In CNS2 KO T cells, Foxp3 expression is lost as cells divide and under inflammatory conditions^{5,6}.
- High IL-2 doses allow CNS2 KO Foxp3⁺ T cells largely maintain Foxp3 expression, whereas low doses lead to reduced Foxp3 expression in CNS2 KO T cells^{6,7}.
- TCR stimulation alone fails to upregulate Foxp3 expression in CNS2 KO T cells⁷.

While these findings firmly established CNS2’s involvement in Foxp3 regulation, they were based entirely on **bulk, endpoint** analyses of endogenous Foxp3 CNS2 KO^{5,6,7}. As such, they could not resolve **which specific cells** exhibit CNS2-dependent activity, nor could they distinguish the **primary effects of CNS2 deletion** from **secondary effects** arising due to reduced Foxp3 protein expression. Moreover, these studies did **not** directly measure the downstream transcriptional effects of TCR and IL-2 signalling within specific cells in which CNS2 is active. Therefore, it remains unresolved how intact CNS2 controls Foxp3 transcription **dynamically** at the

single-cell level—specifically, what types of transcriptional activity CNS2 induces, in which cells, and what gene regulation accompanies its activity.

1.2. Novelty of Single-Cell Dynamics: Identifying specific cells in which CNS2 is actively regulating Foxp3 transcription—and characterizing their transcriptional dynamics—has been beyond the reach of existing methods (p. 4 line 11–24). By contrast, our ML methods enable high-resolution, data-driven identification of CNS2-functional cells, supported by strong performance metrics that surpass manual gating (p. 14 line 7–11. **Figs. 4e–4f**). Through this approach, we have revealed **high-frequency** Foxp3 transcriptional bursts in CNS2-functional cells —dynamics not observed in CNS2-independent Foxp3⁺ cells, which exhibit attenuated and arrested transcription (highlighted in blue in **Fig. 4g–4h**). Importantly, by deleting CNS2 specifically within the Foxp3 Tocky transgene while preserving the endogenous CNS2, we have **effectively dissociate the primary effects of CNS2 deletion from the secondary effects due to attenuated Foxp3 protein levels, overcoming the critical limitation faced by prior studies** that employed existing CNS2 KO strategies.

Collectively, these innovations not only represent a significant advancement beyond previous studies that viewed CNS2 primarily as a maintenance element during cell division but also establish a proof-of-concept for dissecting physiological transcriptional dynamics at the single-cell level (p. 24 line 2–5). Future studies may build on the Tocky-ML tools by integrating single-cell methylation analysis at CpGs in CNS2, and/or chromatin configuration analysis, to further elucidate regulatory mechanisms in vivo (p. 27, line 13 – 18).

1.3. Gene Expression Profiles: The assumption that “CNS2-dependent cells uniformly have higher TCR and IL-2 signalling signatures” is an oversimplification. Our data indicate that CNS2-dependent cells exhibit **nuanced** gene expression profiles downstream of IL-2 and TCR signaling (p. 16 line 6 – p.17 line 2. Specifically, we observed upregulated IL-2R (reflecting IL-2 input and responsiveness) and Nfatc1/2 (TCR downstream), alongside **selective repression** of other key genes such as Stat5, NF-kB, and Nr4a.

While previous studies only indirectly suggested the role of TCR and IL-2 signals in ‘maintaining Foxp3 expression in CNS2 WT cells’, our findings provide a more **refined picture**, showing that **CNS2 operates under fine and unique temporal dynamics of TCR and IL-2 signalling**. We propose that these nuanced dynamics likely involve **periodic** and **brief** TCR and IL-2 inputs (p. 26 line 18–p. 27 line 9).

2. Reviewer’s Comments: *“The validation of the new methods and its comparison with flow cytometry is based on RNA-seq of FACS-sorted cells and comparisons of marker MFIs of feature cells and “other cells”. These validation methods demonstrate that the feature cells can indeed be identified and isolated using flow cytometry. It is unclear whether the “transcriptional dynamics” of CNS2-dependent Foxp3 expression can be discovered by sorting and sequencing Timer+ cells that are only present in WT mice (those that are top left in a timer red vs timer blue plot) vs those only present in KO mice (timer red+ timer blue- cells)”.*

Thank you for acknowledging our methods’ capability to identify feature cells. Below, we clarify how our ML methods enable analysis of CNS2-induced transcription within **WT Tocky**, departing from prior studies:

(i) ML-Assisted Identification of CNS2-Dependent Transcription: In contrast to previous KO studies, our integrated ML approach enables precise identification of cells that represent group-specific properties within the dataset, designated as *‘feature cells’*, which **fundamentally differs from any gating methods** (new **Fig. 1b**, p. 7 line 11–p. 8 line 4). The ConvNet method distinguishes cells where CNS2-dependent transcription is

occurring (**WT features**) from those where transcription is independent of CNS2 activity (**KO features**), both within WT Foxp3 Tocky T cells (Fig. 4g–4h; p. 14 line 13–25).

(ii) **Analysis of WT and KO Features:** Thank you for your constructive comment. We first clarify that analyzing KO feature cells in CNS2 KO Tocky cells does **not** provide meaningful insights, because the endogenous Foxp3 gene—and thus CNS2—remains intact in CNS2 KO Tocky. In contrast, the ConvNet model, which has been **trained to capture the patterns of both WT and CNS2 KO Tocky**, enables identification of KO feature cells within **WT Foxp3 Tocky**. Thus, analyzing **both WT feature cells and KO feature cells from WT Tocky mice** has allowed us to elucidate gene expression dynamics in CNS2-functioning cells. Our new analysis demonstrated that the fractions **R1 and R2 are enriched with KO features, i.e. CNS2-independent cells** (Figs. 5b and 5c, p. 15 line 18–p.16 line 5). These additions have enhanced the comparative analysis in **Figs. 5d and 5e**, highlighting the importance of significant changes in key genes and offering a clear, direct view of gene expression profiles induced by functioning CNS2 (p. 16 line 6–18).

3. Reviewer’s Comment: *“The added thymus vs spleen ageing analysis is performed rather crudely and fails to strengthen the manuscript. Thus, the enthusiasm for publishing this manuscript remains limited. ... The application of the ML methods to characterize the effect of ageing on transcriptional dynamic of Foxp3 appears rudimentary. As mouse age, the composition of thymic CD4+ T cells changes, the observed differences in timer red/timer blue ratio could be explained by the increased presence of mature Treg cells recirculating from the periphery and consequently not actively upregulating Foxp3. This difference in Foxp3 transcriptional dynamics in CD4 T cells cannot be simply attributed to ageing. Thus, the merit of the analysis shown in Figures 6 and 7 is questionable.”*

Response: As already clarified above, our ageing analysis is rigorous both biologically and methodologically, and far from “crude”. Here we specifically address how **recirculation** was accounted for in our study:

3-1. The updated Results (p. 19, line 18–p. 20, line 8) clarifies our rationale for classifying mice under 30 days old as “young.” Given that recirculation of Foxp3⁺ T cells typically begins after 6–7 weeks of age⁸, this threshold ensures that the young thymus is not confounded by recirculated cells. This also aligns with accepted definitions of young adult thymus⁹ and provides balanced groups for ML training (**Suppl. Table 1**).

3-2. The updated Discussion (p. 28 line 15–p. 29 line 9) clarifies that existing tools such as the Rag2p-EGFP reporter are limited due to EGFP downregulation during thymic cell division, which may lead to overestimation of recirculation^{10, 11}. To address this, we propose developing a new tool—Rag2-Tocky—to enable precise, time-resolved analysis of TCR recombination and Foxp3 transcription (p. 29 line 2–7).

3-3. All major biological discoveries from the ageing analysis are **independent of recirculation concerns:**

- a. Our primary finding, highlighted in **Abstract**, is the “thymus-like” features of neonatal splenic T cells, which peaks around **days 3–4 post-birth** as detailed in Discussion (p. 29 line 10–23).
- b. The **updated Discussion** shows that, even within **neonatal thymic** T cells, thymus-specific transcriptional features peak around day 4 post-birth, as demonstrated by our TockyConvNet continuous model (p. 28 line 7–12). In addition, it shows a progressive decline in Foxp3 transcriptional frequency within **aged splenic** T cells (p. 29, line 24–p. 30, line 16), revealing novel senescent effects on Foxp3 transcriptional regulation **in the spleen**.

These findings are all independent of the recirculated T cells in the aged thymus.

References

1. Fontenot, J.D., Dooley, J.L., Farr, A.G. & Rudensky, A.Y. Developmental regulation of Foxp3 expression during ontogeny. *J Exp Med* **202**, 901-906 (2005).
2. Lages, C.S. *et al.* Functional Regulatory T Cells Accumulate in Aged Hosts and Promote Chronic Infectious Disease Reactivation. *The Journal of Immunology* **181**, 1835-1848 (2008).
3. Sharma, S., Dominguez, A.L. & Lustgarten, J. High Accumulation of T Regulatory Cells Prevents the Activation of Immune Responses in Aged Animals. *The Journal of Immunology* **177**, 8348-8355 (2006).
4. Bending, D. *et al.* A temporally dynamic Foxp3 autoregulatory transcriptional circuit controls the effector Treg programme. *The EMBO journal* **37**, e99013 (2018).
5. Zheng, Y. *et al.* Role of conserved non-coding DNA elements in the Foxp3 gene in regulatory T-cell fate. *Nature* **463**, 808-812 (2010).
6. Feng, Y. *et al.* Control of the inheritance of regulatory T cell identity by a cis element in the Foxp3 locus. *Cell* **158**, 749-763 (2014).
7. Li, X., Liang, Y., LeBlanc, M., Benner, C. & Zheng, Y. Function of a Foxp3 cis-element in protecting regulatory T cell identity. *Cell* **158**, 734-748 (2014).
8. Thiault, N. *et al.* Peripheral regulatory T lymphocytes recirculating to the thymus suppress the development of their precursors. *Nat Immunol* **16**, 628-634 (2015).
9. Rowell, J. *et al.* Distinct T-cell receptor (TCR) gene segment usage and MHC-restriction between foetal and adult thymus. *eLife* **13**, RP93493 (2024).
10. Kirberg, J.r., Bosco, N., Deloulme, J.-C., Ceredig, R. & Agenès, F. Peripheral T Lymphocytes Recirculating Back into the Thymus Can Mediate Thymocyte Positive Selection. *The Journal of Immunology* **181**, 1207-1214 (2008).
11. Hale, J.S., Boursalian, T.E., Turk, G.L. & Fink, P.J. Thymic output in aged mice. *Proceedings of the National Academy of Sciences* **103**, 8447-8452 (2006).